# Learning to Reuse Policies in State Evolvable Environments

**Ziqian Zhang** [* 1 2]  **Bohan Yang** [* 1 2]  **Lihe Li** [1 2]  **Yuqi Bian** [1 2]  **Ruiqi Xue** [1 2]  **Feng Chen** [1 2]  **Yi-Chen Li** [1 2]
**Lei Yuan** [1 2 3]  **Yang Yu** [1 2 3]

## Abstract

The policy trained via reinforcement learning (RL) makes decisions based on sensor-derived state features. It is common for state features to evolve for reasons such as periodic sensor maintenance or the addition of new sensors for performance improvement. The deployed policy fails in new state space when state features are unseen during training. Previous work tackles this challenge by training a sensor-invariant policy or generating multiple policies and selecting the appropriate one with limited samples. However, both directions struggle to guarantee the performance when faced with unpredictable evolutions. In this paper, we formalize this problem as state evolvable reinforcement learning (SERL), where the agent is required to mitigate policy degradation after state evolutions without costly exploration. We propose **Lapse** by reusing policies learned from the old state space in two distinct aspects. On one hand, Lapse directly reuses the *robust* old policy by composing it with a learned state reconstruction model to handle vanishing sensors. On the other hand, the behavioral experience from the old policy is reused by Lapse to train a newly adaptive policy through offline learning, better utilizing new sensors. To leverage advantages of both policies in different scenarios, we further propose *automatic ensemble weight adjustment* to effectively aggregate them. Theoretically, we justify that robust policy reuse helps mitigate uncertainty and error from both evolution and reconstruction. Empirically, Lapse achieves a significant performance improvement, outperforming the strongest baseline by about $2\times$ in benchmark environments.

## 1. Introduction

Deep reinforcement learning (RL) has shown tremendous potential in various applications (Wang et al., 2022), such as sequential recommendation systems (Lin et al., 2023) and robotic control (Singh et al., 2022). However, deploying RL agents in the real world faces a major challenge: policies trained under a fixed state feature space often fail to generalize to new, unseen ones in open environments (Zhou, 2022; Yuan et al., 2023). For instance, autonomous driving agents utilize heterogeneous sensor sources for information gathering (Kiran et al., 2021). However, the sensors might wear out due to limited lifespan or require maintenance, and new sensors with different camera views may be deployed to improve performance from richer informatioin (Shukla et al., 2023). Such common scenarios lead to continuous changes (evolutions) in the state space during deployment, requiring agents to adapt to the state evolvale environments.

This presents a critical challenge: adapting to new state spaces demands extensive interactions to learn a policy, which is costly in online deployment. Existing methods learn a sensor-invariant policy that is robust to changes in the state space (Chen et al., 2021; Li et al., 2022; Hansen et al., 2021), or generate multiple source policies beforehand and select the appropriate one using hierarchical high-level policies or value function estimation (Cheng et al., 2020; Yang et al., 2020; Zhang et al., 2022; Chen et al., 2023). However, learning a sensor-invariant policy requires determining which aspects of state features are task-specific and which are shared. This is complex and previous methods often assume prior knowledge about new state spaces (Laskin et al., 2020a; Zhou et al., 2023). Meanwhile, enumerating all possible state spaces is a daunting task, making it impractical to generate sufficiently diverse source policies for reuse (Cheng et al., 2020; Barekatain et al., 2021). Consequently, both directions fail to guarantee the performance when faced with unpredictable state evolutions.

To tackle the challenges posed by evolvable state features, we formalize the problem as state evolvable reinforcement learning (SERL), where unknown evolutions map the old state space to a new one, leading to changes in transition and reward functions as well. To mitigate performance degradation resulted from state space evolutions, we propose

---

[*]Equal contribution [1]National Key Laboratory for Novel Software Technology, Nanjing University [2]School of Artificial Intelligence, Nanjing University [3]Polixir Technologies. Correspondence to: Lei Yuan <yuanl@lamda.nju.edu.cn>, Yang Yu <yuy@nju.edu.cn>.

*Proceedings of the $42^{nd}$ International Conference on Machine Learning*, Vancouver, Canada. PMLR 267, 2025. Copyright 2025 by the author(s).

**Lea**rning to reuse **p**olicies in **s**tate **e**volvable environments (Lapse) by reusing the old policy in two different directions. On one hand, by assuming the prior knowledge of sensors designated for maintenance, agents can simultaneously observe features from old and new state spaces for a short period. Using a state reconstruction model trained with Conditional Generative Adversarial Networks (GANs) (Isola et al., 2017), the old policy can be directly reused by reconstructing the old state features from the new ones, thereby handling issues like vanishing sensors. However, uncertainty in state evolution and reconstruction errors may cause unbounded performance gaps, we minimize discrepancies in action distributions under perturbations, thereby bounding performance degradation. From another perspective, an adaptive policy is trained using the behavioral knowledge carried out by its experience via offline RL, so as to better utilize new sensors. Finally, to benefit from the advantages of two policies in different scenarios, we aggregate them with automatic adjustment of ensemble weights, ensuring better performance as the state space evolves.

Our theoretical study discloses the gap caused by the uncertainty of state evolution and justifies the use of policy robustness. Extensive experiments are performed both in MuJoCo control tasks (Todorov et al., 2012) with vectorial state features and Atari games (Bellemare et al., 2013) with pixel-based images. To evaluate the adaptation capability, we test the policy in environments with multiple unknown state space evolutions. The empirical results showcase a remarkable improvement in our method compared to existing adaptation and transfer approaches, achieving about $2\times$ performance enhancement for state evolvable environments. Our code is available at https://github.com/zzq-bot/Lapse

## 2. Problem Formulation

The standard reinforcement learning (RL) problem (Sutton & Barto, 2018) is formally defined by a Markov decision process (MDP), denoted as $\mathcal{M} = \{\mathcal{S}, \mathcal{A}, \mathcal{P}, R, \gamma\}$. Here $\mathcal{S}$ and $\mathcal{A}$ represent state and action space, respectively. $\mathcal{P} : \mathcal{S} \times \mathcal{A} \times S \to [0, 1]$ is the transition function, $R : \mathcal{S} \times \mathcal{A} \to [0, R_{\max}]$ is the reward function, and $\gamma \in [0, 1)$ signifies the discount factor. At each time step $t$, the agent perceives the state $s_t$ and decides an action based on its policy $a_t \sim \pi(\cdot|s_t)$. This leads to the next state $s_{t+1} \sim \mathcal{P}(\cdot|s_t, a_t)$ and a reward $R(s_t, a_t)$. The primary objective is to find a policy that maximizes the expected discounted return $J(\pi) = \mathbb{E}[\sum_{t=0}^{\infty} \gamma^t R(s_t, a_t)|s_0, \pi, \mathcal{P}]$.

In this work, we consider an environment where the state space evolves over time. This scenario is formulated as a state evolvable reinforcement learning (SERL) problem: $\{(\mathcal{M}_n, f_n, D_n)\}_{n=0}^N$. Each evolving MDP is denoted by $\mathcal{M}_n = \{\mathcal{S}_n, \mathcal{A}, \mathcal{P}_n, R_n, \gamma\}$, and the unknown multivalued

mapping $f_n : \mathcal{S}_n \to \Delta(\mathcal{S}_{n+1})$ governs how the state space evolves from $\mathcal{S}_n$ to $\mathcal{S}_{n+1}$, where $\Delta(\cdot)$ is the probability simplex. $N \in \mathbb{Z}_+ \cup \{+\infty\}$ is the number of evolutions. The goal of SERL is to maximize the adaptation performance of policy $\pi_n$ during deployment, without the need for costly trial-and-error. Given that the evolution of the state space is driven by changes in sensors, we can collect limited experiences $D_n = \{(s_n, s_{n+1}, a, r, s'_n, s'_{n+1})\}$ via the policy $\pi_n$. We make a mild assumption that prior knowledge of which sensors will undergo maintenance and which new sensors will be deployed is available. This allows agents to perceive paired states $(s_t, s_{t+1})$ in a *short* period by simply masking the corresponding state features.

As our focus is on the evolution of the state space, the reward and transition functions remain consistent. However, due to unpredictable noise, $f_n$ may be stochastic, non-bijective, and multivalued. To manage this uncertainty, we define a Bayesian-style inverse $f_n^{-1} : \mathcal{S}_{n+1} \to \Delta(S_n)$, where $f_n^{-1}(s_n|s_{n+1}) = \frac{f_n(s_{n+1}|s_n)}{\sum_{\tilde{s}_n} f_n(s_{n+1}|\tilde{s}_n)}$ and quantify the uncertainty by defining $\epsilon_R - \epsilon_P$ consistency in Section 3.4.

**Definition 2.1.** Let $\mathcal{M}_n = \{\mathcal{S}_n, \mathcal{A}, \mathcal{P}_n, R_n, \gamma\}$ be an evolving MDP, with evolution $f_n : \mathcal{S}_n \to \Delta(\mathcal{S}_{n+1})$ and inverse $f_n^{-1} : \mathcal{S}_{n+1} \to \Delta(\mathcal{S}_n)$. We define $\mathcal{M}_{n+1} = \{\mathcal{S}_{n+1}, \mathcal{A}, \mathcal{P}_{n+1}, R_{n+1}, \gamma\}$, where $\mathcal{S}_{n+1} = \{s_{n+1} : s_{n+1} \sim f(\cdot|s_n), s_n \in \mathcal{S}_n\}$, $R_{n+1}(s_{n+1}, a) = \mathbb{E}_{s_n \sim f_n^{-1}(\cdot|s_{n+1})}[R_n(s_n, a)]$, and $\mathcal{P}_{n+1}(s'_{n+1}|s_{n+1}, a) = \mathbb{E}_{s_n, s'_n \sim f_n^{-1}(\cdot|s_{n+1}), f_n^{-1}(\cdot|s'_{n+1})}[\mathcal{P}_n(s'_n|s_n, a)]$.

## 3. Method

In this section, we will describe our proposed **Lea**rning to reuse **p**olicies in **s**tate **e**volvable environments (**Lapse**). As shown in Figure 1, Lapse learns to reuse the old policy in two directions. On one hand, we learn a state reconstruction model and directly reuse the old robust policy. On the other hand, behavior knowledge contained in limited experience is reused to learn a new policy with robustness regularization. Moreover, we aggregate them with automatic adjusted ensemble weight. Finally, we give a theoretical analysis of how uncertainty of evolution brings about performance degradation and a bound of the performance gap.

### 3.1. Robust Policy Reuse with State Reconstruction

When a policy $\pi_n$, effective in $\mathcal{M}_n$, encounters $\mathcal{M}_{n+1}$, it fails due to the evolution of the state space. However, if old state features $s_n$ can be reconstructed from new $s_{n+1}$, we can reuse $\pi_n$ by composing it with the learned reconstruction model. Leveraging paired state features $(s_n, s_{n+1})$, we can utilize supervised paired GANs (Isola et al., 2017) to learn a state reconstruction model $g_n : \mathcal{S}_{n+1} \to \Delta(\mathcal{S}_n)$, alongside a discriminator $d_n : \mathcal{S}_{n+1} \times \mathcal{S}_n \to [0, 1]$. The

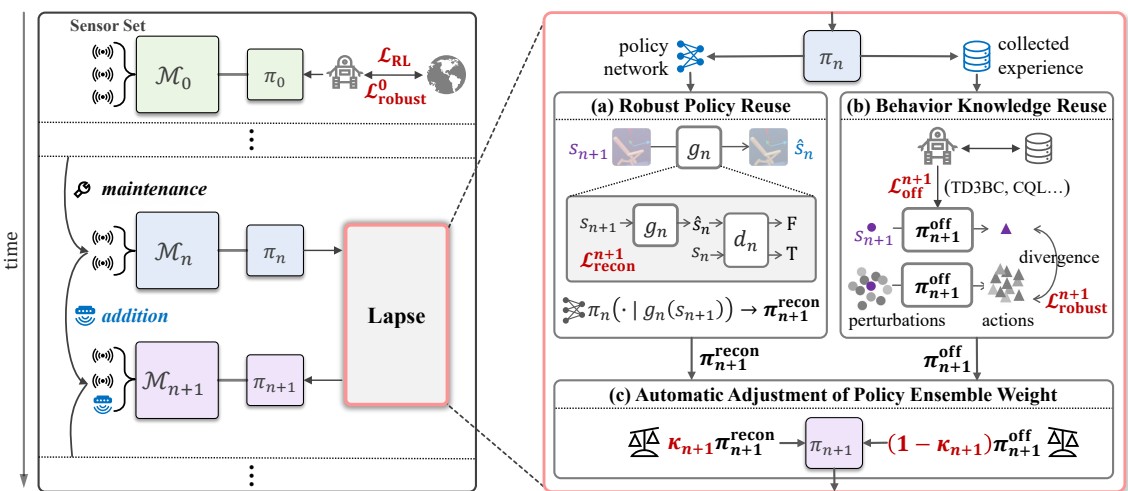

*Figure 1.* The overall framework of SERL and Lapse

objective of the conditional GAN can be expressed as:

$$\mathcal{L}_{\text{GAN}}^{n+1} = \mathbb{E}[\log d_n(s_{n+1}, s_n) + \log(1 - d_n(s_{n+1}, \hat{s}_n))], \quad (1)$$

where $\hat{s}_n \sim g_n(s_{n+1})$. $g_n$ aims to minimize this objective against $d_n$ which seeks to maximize it, i.e., $g_n^* = \arg\min_{g_n} \max_{d_n} \mathcal{L}_{\text{GAN}}^{n+1}$. Additionally, we impose an $l_p$-based reconstruction loss to ensure the generated $\hat{s}_n$ closely matches $s_n$:

$$\mathcal{L}_{\text{Lp}}^{n+1} = \mathbb{E}_{s_n, s_{n+1}, \hat{s}_n \sim g_n(\cdot|s_{n+1})}[\|s_n - \hat{s}_n\|_p], \quad (2)$$

where $p$ is a predefined parameter. Concretely, we set $p = 1$ to encourage less blurring in pixel-based environment and $p = 2$ in vectorial environments. The overall reconstruction objective is thus defined as follows:

$$\mathcal{L}_{\text{recon}}^{n+1} = \mathcal{L}_{\text{GAN}}^{n+1} + \lambda \mathcal{L}_{\text{Lp}}^{n+1}, \quad (3)$$

where $\lambda$ is a tuneable hyper-parameter. It is also feasible to instantiate the reconstruction model by diffusion networks, but we here select GAN for its faster inference time considering time-delay during deployment.

By composing the reconstruction model with the old policy, we obtain the policy with $a \sim \pi_{n+1}^{\text{recon}}(\cdot|s_{n+1}) := (\pi_n \circ g_n)(\cdot|s_{n+1}) = \pi_n(\cdot|\hat{s}_n)$, where $\hat{s}_n \sim g(\cdot|s_{n+1})$. Since $g$ is trained to approximate $f_n^{-1}$, we show in Section 3.4 that this reconstruction-based policy can transfer effectively to $\mathcal{M}_{n+1}$. In practice, however, limited data coverage during the short period and reliance on a fixed behavior policy can hinder $g_n$'s ability to generalize to unseen states. Additionally, standard RL policies tend to be sensitive to perturbations of their inputs (Zhang et al., 2020a), so the reconstruction error may lead to erroneous actions, that drive the agent into unobserved states, further compounding the reconstruction error.

We mitigate these issues by learning a *robust* policy that reduces the impact of small input perturbations on action

distributions. Concretely, we incorporate a robustness regularization when training $\pi_n$ under $\mathcal{M}_n$:

$$\mathcal{L}_{\text{robust}}^n = \mathbb{E}\left[\max_{\hat{s}_n \in T_n^\epsilon(s_n)} D_{\text{TV}}(\pi_n(\cdot|s_n), \pi_n(\cdot|\hat{s}_n))\right], \quad (4)$$

where $T_n^\epsilon(s_n) = \{s_n' | \|s_n' - s_n\|_2 \leq \epsilon, s_n' \in \mathcal{S}_n\}$ is a $\epsilon$-neighborhood of $s_n$. With the reconstruction model only, we could define $\pi_n = \pi_0 \circ g_0 \circ ... \circ g_{n-1}$. Because $\pi_0$ is trained in $\mathcal{M}_0$ before deployment, when trial-and-error is acceptable, we can use any robust RL algorithm by optimizing:

$$\mathcal{L}_0 = \mathcal{L}_{\text{RL}} + \alpha_{\text{robust}} \mathcal{L}_{\text{robust}}^0. \quad (5)$$

Here, $\mathcal{L}_{\text{RL}}$ is the standard RL objective (e.g., TD-target for DQN (Mnih et al., 2013)), and $\alpha_{\text{robust}}$ weights the robustness term $\mathcal{L}_{\text{robust}}^0$. Any robust RL method (e.g., RADIAL-DQN (Oikarinen et al., 2021), Wocar-PPO (Liang et al., 2022)) can be employed for this step. We defer the theoretical performance-gap analysis under robustness guarantees to Section 3.4. When extending Lapse to environments with different state or action spaces, we simply adapt the method to each environment's standard RL backbone. Further implementation details can be found in Appendix B.

## 3.2. Offline Policy Learning with Knowledge Reuse

Despite the effectiveness of robust policy reuse when old sensors vanish for maintenance or power failure, it fails to take advantage of the information provided by newly added sensors. Meanwhile, the reconstruction error from the composite function $\hat{g}_n := g_0 \circ g_1 ... \circ g_n$ will accumulate as the state evolution continues. Relying solely on direct policy reuse will lead to an increasing discrepancy in the action distribution $\mathbb{E}[D_{\text{TV}}(\pi_0(\cdot|s_0), \pi_0 \circ \hat{g}_n(\cdot|s_{n+1}))]$, and the compounding error from the composition of policy and reconstruction function becomes problematic again.

To mitigate the issues, we propose to complement policy reuse with the reuse of behavioral knowledge to further

promote adaptation in the new MDP. Although the reconstruction policy may deteriorate over time, its performance is stable for the earlier stages, suggesting a decent quality of offline dataset $D_n = \{(s_n, s_{n+1}, a, r, s'_n, s'_{n+1})\}$. A straightforward method is to apply behavior cloning which attempts to simply clone the actions observed in the dataset, but it also fails to leverage the additional information gained from the newly added sensors. Therefore, we turn to offline RL for policy learning, as suggested by Kumar et al. (2022).

For environments with vectorial state features and continuous action space such as Mujoco, we employ the TD3+BC (Fujimoto & Gu, 2021) to learn the offline policy $\pi_{n+1}^{\text{off}}$. The loss function is optimized as follows:

$$\hat{\mathcal{L}}_{\text{off}}^{n+1} = -\mathbb{E}_{(s_{n+1}, a) \sim D_n}\big[ -(\pi_{n+1}^{\text{off}}(s_{n+1}) - a)^2 + \beta_{n+1} Q_{n+1}(s_{n+1}, \pi_{n+1}^{\text{off}}(s_{n+1}))\big], \quad (6)$$

where $Q_{n+1}$ is learned via the TD3, and $\beta_{n+1}$ is a coefficient controlling conservatism. We schedule the coefficient dynamically with state feature evolution, setting $\beta_{n+1} = \beta_{\max}(1 - \exp(-\tau \cdot n))$, where $\tau$ is a decay coefficient. Intuitively, when reconstruction error is small (in early stages), we rely more on imitating the behavior policy. As uncertainty grows, we tilt toward more optimistic updates. To further enhance robustness, we incorporate a robust regularization term:

$$\mathcal{L}_{\text{off}}^{n+1} = \hat{\mathcal{L}}_{\text{off}}^{n+1} + \alpha_{\text{robust}}^{n+1} \mathcal{L}_{\text{robust}}^{n+1}, \quad (7)$$

where $\alpha_{\text{robust}}^{n+1}$ balances the regularization, and $\mathcal{L}_{\text{robust}}^{n+1}$ is given in Equation (4). Different from online training, we implement the robust regularization via function smoothing based on robust offline RL proposed by RORL (Yang et al., 2022). For environments with pixel-based image inputs and discrete action space, we adopt suitable offline RL backbones tailored to such settings. Detailed architectural and algorithmic choices are provided in Appendix C.

### 3.3. Policy Aggregation with Automatic Adjustment of the Ensemble Weight

The two distinct ways of policy reuse are proficient in different scenarios. Finally, to leverage their advantages, we combine them via an ensemble whose weight is adjusted automatically (Lee et al., 2021). Specifically, for $\mathcal{M}_{n+1}$, the policy $\pi_{n+1}$ is recurrently defined as:

$$\pi_{n+1} = \kappa_{n+1}\pi_{n+1}^{\text{recon}} + (1 - \kappa_{n+1})\pi_{n+1}^{\text{off}}, \qquad n \geq 0, \quad (8)$$

where $\pi_{n+1}^{\text{recon}} = \pi_n \circ g_n$, $\pi_0$, $g_n$ and $\pi_{n+1}^{\text{off}}$ are learned via Equations (5), (3), and (7), respectively. We set

$$\kappa_{n+1} = \frac{J_n(\pi_n)}{J_0(\pi_0)} \cdot \frac{D(\pi_n, \pi_{n+1}^{\text{off}})}{D(\pi_n, \pi_{n+1}^{\text{recon}}) + D(\pi_n, \pi_{n+1}^{\text{off}})}, \quad (9)$$

where $J_n(\pi_n)$ is the expected return of policy $\pi_n$ under $\mathcal{M}_n$, and $D(p, q)$ is a divergence metric (e.g., KL divergence). If $\pi_n$ performs comparably to $\pi_0$, then $\kappa_{n+1}$ emphasizes whichever policy is closest to $\pi_n$. Otherwise, the newly adaptive policy, which is expected to exceed the behavior policy via offline RL, takes precedence.

To summarize the overall training and testing processes under the SERL framework, Lapse first learns a robust initial policy $\pi_0$ via Equation. (5). During deployment, $\pi_n$ makes decisions in $\mathcal{M}_n$. Prior to evolving into $\mathcal{M}_{n+1}$, $\pi_n$ could perceive both state feature from $\mathcal{S}_n$ and $\mathcal{S}_{n+1}$ for a short period and collects limited amount of experience. The reconstruction model and the offline adaptive policy are then trained to obtain $\pi_{n+1}^{\text{recon}} = \pi_n \circ g_n$ and $\pi_{n+1}^{\text{off}}$, respectively. Our aggregation method automatically adjusts the ensemble weight $\kappa_{n+1}$ of $\pi_{n+1}$, facilitating transfer to $\mathcal{M}_{n+1}$. For memory efficiency, one may also prune past policies with negligible performance loss. Detailed pseudocode is provided in Appendix C.

### 3.4. Theoretical Analysis

We now analyze how a robust policy, combined with a learned state reconstruction model, can limit performance degradation under evolvable state space. To characterize the uncertainty introduced by evolution $f_n$, we first define $\epsilon_R - \epsilon_P$ consistency:

**Definition 3.1.** Given $\mathcal{M}_n$ and $f_n$, we say $f_n$ is $\epsilon_R - \epsilon_P$ consistent, if for any $s_n \in S_n, s_{n+1} \in S_{n+1}, a \in \mathcal{A}$, with $f_n(s_{n+1}|s_n) > 0$, the following holds:

$$\Big|R_n(s_n, a) - \sum_{\tilde{s}_n} f_n^{-1}(\tilde{s}_n|s_{n+1})R_n(\tilde{s}_n, a)\Big| \leq \epsilon_R,$$

$$||\mathcal{P}_n(s_n, a) - \mathcal{F}_n^{-1}\mathcal{P}_{n+1}(s_{n+1}, a)||_1 \leq \epsilon_P.$$

Here, $\mathcal{P}_n(s_n, a) \in \mathbb{R}^{|\mathcal{S}_n|}$ is the transition distribution for $(s_n, a)$, and $\mathcal{F}_n^{-1}$ is a $|\mathcal{S}_n| \times |\mathcal{S}_{n+1}|$ matrix with $\mathcal{F}_n^{-1}(s_n, s_{n+1}) = f_n^{-1}(s_n|s_{n+1})$. Thus, $\mathcal{F}_n^{-1}\mathcal{P}_{n+1}(s_{n+1}, a)$ collapses the transition distribution over $\mathcal{S}_{n+1}$ to one over $\mathcal{S}_n$.

Intuitively, $\epsilon_R - \epsilon_P$ consistency controls the discrepancy of reward and transition functions in $\mathcal{M}_n$ and $\mathcal{M}_{n+1}$, thereby bounding the uncertainty of the evolution $f_n$ and $f_n^{-1}$.

Our starting point is the theoretical effectiveness of the state reconstruction model. If we have direct access to the inverse $f_n^{-1}$ of the evolution $f_n$, we can immediately reuse $\pi_n$ by composing it with the $f_n^{-1}$ to get $\pi_n \circ f_n^{-1}$. The following proposition shows that the performance loss in this scenario only results from the $\epsilon_R - \epsilon_P$ consistency of $f_n$.

**Proposition 3.2.** *Let $f_n$ be $\epsilon_R - \epsilon_P$ consistent, and $f_n^{-1}$ be its inverse. Suppose $f_n^{-1}(s_{n,0}|s_{n+1,0}) = 1$ for the initial state $s_{n,0}$ and $s_{n+1,0}$ in $\mathcal{M}_n$ and $\mathcal{M}_{n+1}$, respectively.*

$J_n(\pi_n) := \mathbb{E}[\sum_{t=0}^{\infty} \gamma^t r_{n,t} | \pi_n, s_{n,0}, \mathcal{P}_n]$. *When applying $\pi_n \circ f_n^{-1}$ in $\mathcal{M}_{n+1}$, the performance gap is bounded by*

$$|J_{n+1}(\pi_n \circ f_n^{-1}) - J_n(\pi_n)| \leq \frac{\epsilon_R}{1 - \gamma} + \frac{\gamma \epsilon_P R_{\max}}{2(1 - \gamma)^2}.$$

The full proof can be found in Appendix A. We disclose the policy performance gap when encapsulating $\pi_n$ with the oracle inverse of evolution $f_n^{-1}$. This quantifies how the uncertainty in the evolution, as captured by $\epsilon_R$ and $\epsilon_P$, affects the performance under a infinite-horizon setting.

In practice, we only have access to a learned reconstruction function $g_n$ that approximates $f_n^{-1}$. Because the data $D_n$ come from trajectories with specific distributions, compounding errors may occur if $g_n$ is inaccurate in off-distribution states. The next proposition shows that a robust policy $\pi_n$ and help mitigate such errors.

**Proposition 3.3.** *Let $f_n^{-1}$ be the inverse of evolution $f_n$ that transforms $\mathcal{M}_n$ into $\mathcal{M}_{n+1}$. Suppose we have both the policy $\pi_n$ and the reconstruction model $g_n$. If*

$$\mathbb{E}\left[D_{TV}(\pi_n \circ f_n^{-1}(\cdot | s^{n+1}), \pi_n \circ g_n(\cdot | s^{n+1}))\right] \leq \eta, \quad (10)$$

*where $D_{TV}(p, q) = \frac{1}{2} \sum_x |p(x) - q(x)| \in [0, 1]$ denotes total variation divergence for discrete probability distribution $p, q$, then:*

$$|J_{n+1}(\pi_n \circ f_n^{-1}) - J_{n+1}(\pi_n \circ g_n)| \leq \frac{2R_{\max}}{(1 - \gamma)^2} \eta.$$

*Moreover, if $f_n$ is $\epsilon_R - \epsilon_P$ consistent, it follows that*

$$|J_n(\pi_n) - J_{n+1}(\pi_n \circ g_n)| \leq \frac{\epsilon_R}{1 - \gamma} + \frac{(\gamma \epsilon_P + 4\eta) R_{\max}}{2(1 - \gamma)^2}.$$

The intuition behind Proposition 3.3 is that if the old policy $\pi_n$ is robust enough to restrict the differences in action distributions under perturbations, then composition with a slightly inaccurate $g_n$ causes limited performance degradation. The detailed proofs are available in Appendix A. We also validate theoretical results through experiments in Section 4.3. Accordingly, when learning $\pi_n$ using dataset $D_{n-1}$, or $\pi_0$ under $\mathcal{M}_0$, we could focus on minimizing $\mathbb{E}[D_{TV}(\pi_n(\cdot | s_n), \pi_n(\cdot | \hat{s}_n))]$, where $\hat{s}_n \sim g_n(\cdot | s_{n+1})$. While we fail to obtain $g_n$ beforehand, we can assume that the reconstructed state $\hat{s}_n$ is bounded by $T_n^\epsilon(s_n) = \{s'_n | \|s'_n - s_n\|_2 \leq \epsilon, s'_n \in \mathcal{S}_n\}$, as $g_n$ is optimized to minimize reconstruction errors. This motivates the robust regularization in Equation (4), as discussed in Section 3.1.

# 4. Experiments

In this section, we present our experimental analysis conducted across eight diverse tasks, including continuous control tasks from Mujoco, featuring vectorial state features, and Atari games with pixel-based image inputs and discrete action spaces. Our experiments are designed to address three critical questions: (1) Whether Lapse achieves superior adaptation capabilities in state evolvable environments compared to existing methods (Section 4.2)? (2) How does the learning process of Lapse proceed in evolving stages, assessing its adaptability (Section 4.3)? (3) What contributions do the different components and hyper-parameters of Lapse make to its overall performance (Section 4.4)?

For a comprehensive evaluation, Lapse is compared against multiple baselines. All results are averaged over five random seeds and are presented with their corresponding standard deviations. Detailed descriptions of the experimental setups, including environmental conditions and network architecture parameters, are provided in the Appendix C.

## 4.1. Baselines and Environments

To thoroughly assess the performance of Lapse, we compare it against the following adaptation baselines: (1) **RL-GAN** (Gamrian & Goldberg, 2019) simply employs a feature reconstruction model to reuse the old policy, but without robust training. (2) **LUSR** (Xing et al., 2021) focuses on extracting disentangled state representations and trains agents in the latent space. (3) **PAD** (Hansen et al., 2021) optimizes RL and self-supervised objectives in the initial MDP ($\mathcal{M}_0$) and adapts by fine-tuning the representation using self-supervised signals. (4) **Offline** learns a new policy for each $\mathcal{M}_{n+1}$ using data $D_n$, employing offline RL. (5) **Few-shot Policy Transfer (FPT)** (Shukla et al., 2023) reuses old policies to guide the transfer of policies under new environments. (6) **CUP** (Zhang et al., 2022) generates multiple source policies in the initial MDP (using randomized domains) and selects the appropriate one based on critic value, which then guides the learning of the new policy.

For evaluation, we select four tasks from the Gym MuJoCo suite (Todorov et al., 2012): `Ant`, `HalfCheetah`, `Hopper`, and `Walker`. In these tasks, agents receive vectorial state features and output continuous actions. We utilize PPO (Schulman et al., 2017) as the backbone and train the initial policy ($\pi_0$) via Wocar (Liang et al., 2022). In the context of Atari games (Bellemare et al., 2013), which involve pixel-based image inputs and discrete action spaces, we employ DQN (Mnih et al., 2013) as the backbone to train the robust RADIAL agent (Oikarinen et al., 2021) $\pi_0$ for games including `BankHeist`, `Freeway`, `Pong`, and `RoadRunner`, following the setup outlined in Oikarinen et al. (2021). The specifics of the state space evolution process in these environments are detailed and illustrated in Appendix B. To simulate the addition and removal of sensors, we split pixel-based images into multiple patches. The addition or reduction of patches or dimensions is then applied. Furthermore, we introduce more complex evolu-

tions including the replacement of physical units, rotation of view, introduction of distracting objects and arbitrary linear mapping. Noise is added to simulate the uncertainty of the evolution. We restrict the number of trajectories in the dataset $D_n$ to 10 and 15 in Mujoco and Atari, respectively.

## 4.2. Competitive Results

**Overall Adaptation Performance** For a fair evaluation of adaptation performance, we analyze all methods across five evolving stages with various orders of evolutions. The performance is normalized against the initial stage return: $100 \cdot \frac{1}{5} \sum_{n=1}^{5} \frac{J_n(\pi_n)}{J_0(\pi_0)}$, with 10 episodes per stage to estimate $J_n(\pi_n)$. This metric reflects how well each method retains performance relative to its initial stage, demostrating the agent's adaptability.

Table 1 reveals that all algorithms experience performance degradation as the state space evolves, underscoring the need for specialized strategies. Domain adaptation methods like RL-GAN and LUSR, which rely on substantial training data to generalize or translate states, struggle because limited samples of $D_n$, exhibiting low adaptability. FPT shows only marginal gains compared to RL-GAN, indicating that few-shot fine-tuning is not always sufficient. PAD, despite learning dynamic-informed intermediate representations through self-supervision, fails to cope with state space evolution. It learns latent features from $\mathcal{M}_0$ alone and it will inevitably be affected by domain specificity, verifying the difficulty to learn a sensor-invariant policy. In addition, PAD fine-tunes the features via limited data with a shallow distribution collected during a short period, this leads to ineffective adaptation. Offline demonstrates decent adaptability in tasks like Ant and Hopper and is on par with Lapse in these scenarios. However, its capability diminishes in Atari games with high-dimensional inputs and complex evolution mappings. CUP shows promise by guiding source policy selection through critic value estimation. However, it is prone to performance degradation when the divergence in state space between the source and target environments becomes too large. This highlights the limitation of previous policy reuse approaches: they are primarily suited for transfer learning and rely on costly trial-and-error to handle significant state space changes. In contrast, Lapse stands out, consistently delivering the best adaptation performance across all tasks. It nearly doubles the effectiveness compared to the strongest baseline in average, evidencing the superiority and efficiency of our proposed method.

**Continuous Adaptation Capability** Figure 2 illustrates the continuous adaptation of various methods to the evolvable state space by comparing the performance over 5 different stages. Although all methods exhibit increased performance degradation over successive stages, Lapse maintains

control over this decline. While RL-GAN's performance deteriorates gradually or breaks down at the first stage, Lapse effectively reuses the old policy, due to the application of the enhanced robustness. LUSR, PAD, and FPT are constrained by the performance of the last stage's policy, yet Lapse transcends these limitations, especially notable in the final stage of HalfCheetah, where it even gains performance improvement in the last stage. This is attributed to the potential of offline adaptive policy learning. Offline and CUP, despite observing similar trends, cannot match Lapse's performance, largely due to their absence of a policy aggregation mechanism with dynamically adjustable ensemble weights. The complete result is provided in Appendix D

## 4.3. Analysis of Policy Reuse

**Validation of Theoretical Analysis** Section 3.4 provides a theoretical discussion on the robustness of policy reuse through state reconstruction. To further validate the theoretical results, we demonstrate the detailed learning process for a single evolving stage in Figure 3. We compare a robust policy and a vanilla policy, each coupled with a reconstruction model trained in the same way. Both the reconstruction models' training losses, depicted by the dotted lines, decrease and stabilize rapidly, indicating minimal reconstruction errors of the old states. This allows the robust policy to be effectively reused in the new state space, delivering satisfactory performance as shown by the red solid line. In contrast, the vanilla policy without prior robust training breaks down, even with a well-trained state reconstruction model. To quantify the total variation divergence $\mathbb{E}[D_{TV}(\pi_n(\cdot|f_n^{-1}(s^{n+1})), \pi_n(\cdot|g_n(s^{n+1})))]$, we measure the action differences produced by the policy when perceiving real and reconstructed states under the testing environment. The blue dash-dotted line that represents the action difference of the vanilla policy has illustrated that the performance gap is mainly caused by the distribution shift and compounding errors. Conversely, a robust policy is capable of mitigating these issues by restricting action distribution discrepancies under perturbations beforehand, thus being reused effectively.

**Analysis of the Learning Process** We further analyze the learning process of Lapse in HalfCheetah, where the reconstruction model and the policy are updated for 10K steps in each evolving stage, as depicted in Figure 4. First, the reconstruction policy $\pi_{n+1}^{recon}$ exhibits high efficacy in early stages, but declines over repeated evolutions. The compounding reconstruction error inevitably accumulates and erodes the initial policy's robustness with the increase of the evolving stages, widening the performance gap. Furthermore, even with an optimal state reconstruction model, the reconstruction policy $\pi_{n+1}^{recon}$ fails to surpass the old policy $\pi_n$. The offline adaptive policy, $\pi_{n+1}^{off}$ shows better long-

*Table 1.* Average test return ± std across various tasks. The highest performance in each row is emphasized in bold. Results are averaged over 5 evolving stages, 10 episodes per stage, and 5 distinct seeds for robustness. Values are normalized against the initial stage return, indicating the performance retained relative to the initial stage.

| Task | RL-GAN | LUSR | PAD | Offline | FPT | CUP | Lapse |
|------|--------|------|-----|---------|-----|-----|-------|
| Ant | $28.10 \pm 3.29$ | $39.57 \pm 1.90$ | $24.47 \pm 1.45$ | $82.63 \pm 4.95$ | $38.39 \pm 5.31$ | $69.83 \pm 9.57$ | $\mathbf{84.02} \pm 2.69$ |
| HalfCheetah | $12.26 \pm 3.06$ | $14.55 \pm 3.35$ | $28.31 \pm 1.07$ | $15.01 \pm 2.71$ | $13.13 \pm 4.36$ | $12.01 \pm 2.02$ | $\mathbf{82.56} \pm 4.48$ |
| Hopper | $16.43 \pm 5.96$ | $15.42 \pm 6.40$ | $34.02 \pm 3.73$ | $74.16 \pm 9.33$ | $78.94 \pm 23.18$ | $66.62 \pm 20.39$ | $\mathbf{98.89} \pm 3.09$ |
| Walker | $3.03 \pm 1.29$ | $4.38 \pm 1.75$ | $10.58 \pm 6.61$ | $18.91 \pm 6.27$ | $19.94 \pm 2.10$ | $13.42 \pm 4.59$ | $\mathbf{83.85} \pm 7.49$ |
| BankHeist | $25.38 \pm 10.58$ | $0.00 \pm 0.00$ | $41.05 \pm 10.43$ | $61.07 \pm 37.43$ | $20.79 \pm 14.78$ | $0.00 \pm 0.00$ | $\mathbf{95.34} \pm 0.89$ |
| Freeway | $43.86 \pm 16.26$ | $61.64 \pm 1.22$ | $78.86 \pm 3.46$ | $71.85 \pm 4.47$ | $78.10 \pm 5.77$ | $48.82 \pm 24.74$ | $\mathbf{98.32} \pm 1.34$ |
| Pong | $37.41 \pm 9.68$ | $0.06 \pm 0.08$ | $6.54 \pm 4.55$ | $33.56 \pm 18.06$ | $33.56 \pm 8.33$ | $0.11 \pm 0.23$ | $\mathbf{100.0} \pm 0.00$ |
| RoadRunner | $14.66 \pm 4.82$ | $2.33 \pm 0.08$ | $28.67 \pm 11.28$ | $6.92 \pm 4.3.0$ | $24.46 \pm 9.00$ | $1.25 \pm 1.04$ | $\mathbf{85.65} \pm 9.33$ |
| Overall | 22.64 | 17.24 | 31.56 | 45.51 | 38.41 | 26.51 | **91.09** |

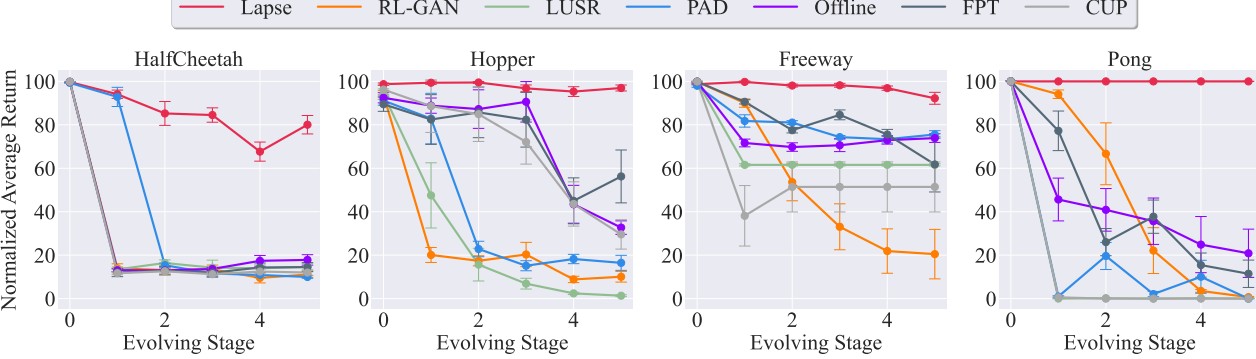

*Figure 2.* The adaptation performance of the learned policy as the state space evolves

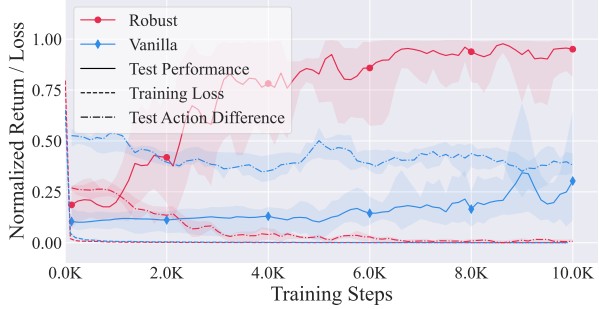

*Figure 3.* Comparison of training loss, test return, and test action difference when reusing robust policies versus reusing vanilla policies through state reconstruction. The type of policy (robust and vanilla) is indicated by color, and different metrics (performance, loss and action difference) are represented by line styles.

term stability and can potentially exceed the performance of a sub-optimal old policy and better utilize new sensors. To benefit from both policies, we aggregate them via an automatically adjusting ensemble weight $\kappa_{n+1}$, which reflects the dependency on the reconstruction policy. As is shown by the dash-dotted line, $\kappa_{n+1}$ initially favors the reconstruction policy due to its closeness to the well-performed old policy. As the state space evolves, $\kappa_{n+1}$ is reduced to harness the

strengths of offline RL based on Equation (9), even without prior knowledge of the individual test performances of the two policies. Incorporating these mechanisms, Lapse realizes superior adaptation performance, outpacing standard approaches. Meanwhile, we illustrate how the two aspects of policy reuse contribute effectively in different scenarios. When the sensor on the backward shin of the HalfCheetah robot wears out, our method learns to reconstruct the missing information from available features, enabling the reuse of the robust policy network. As the reconstruction loss converges, the robust policy can be effectively deployed. Moreover, we visualize the saliency map of offline adaptive policy $\pi_n^{\text{off}}$'s state features via SmoothGrad (Smilkov et al., 2017). The added sensor on the forward shin provides additional state features. The offline policy not only reuses behavior knowledge from the old policy but also leverages the information for improved decision-making.

### 4.4. Ablation and Parameter Sensitivity Studies

**Ablation Study** To dissect the impact of each component within Lapse, we execute ablation studies on the Pong task, as depicted in Figure 5(a). This analysis uncovers their contributions to Lapse in evolvable state space. First, *W/o recon*

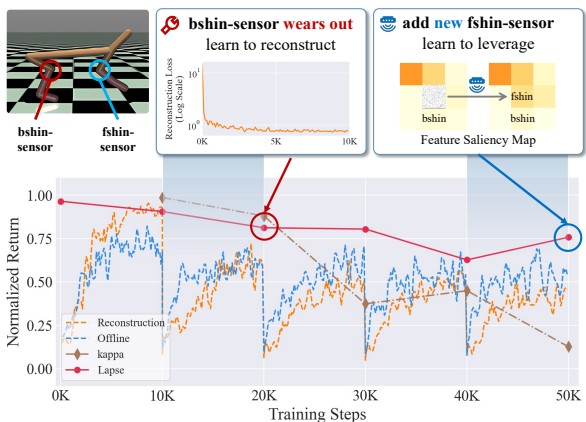

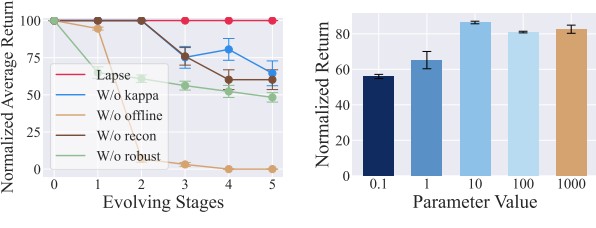

*Figure 4.* The learning process of Lapse in HalfCheetah environment. Displayed are the test returns for the reconstruction policy, the offline adaptive policy, and the combined Lapse policy. Also plotted the ensemble weight ($\kappa_{n+1}$), illustrating its dynamic adjustment throughout evolution. When sensor placed on the backward shin wears out, the model is optimized to reconstruct the missing feature. When a new sensor is added to the forward shin, the policy can utilize new feature, as shown in saliency map.

**(a) Ablation study**

**(b) Parameter sensitivity study**

*Figure 5.* Ablation and parameter sensitivity studies on Pong and Ant, respectively.

and *W/o offline* relies solely on $\pi_{n+1}^{\text{off}}$ and $\pi_{n+1}^{\text{recon}}$, respectively. Both of their adaptation performance declines in later stages, especially *W/o offline*, emphasizing the necessity of the policy aggregation mechanism. Moreover, *W/o robust* learns policies without robustness regularization and this results in unbounded performance gap for the reconstruction policy. Its inferior performance compared to *W/o recon* also suggests that robust training aids in offline learning. *W/o kappa* ensembles two policies with a static weight $\kappa_{n+1} = 0.5$. It suffers the least performance degradation among all ablations but still performs worse than Lapse, manifesting the robustness of the policy aggregation and the benefit of the automatic ensemble weight adjustment.

**Parameter Sensitivity Study**   We probe the influence of hyperparameters on adaptation efficacy in the Ant. Here we examine variations in $\lambda$, which balances the GAN objective and the fidelity of state reconstruction. An excessively low $\lambda$ leads to a misalignment between state spaces, while an overly high $\lambda$ could hinder the ability to fool the discriminator. We find that $\lambda = 10$ is the best choice in Figure 5(b). Additional experimens are detailed in the Appendix D.

## 5. Related Work

**Domain Randomization and Domain Adaptation**   Domain randomization is a widely used technique for enhancing policy generalization (Laskin et al., 2020a; Peng et al., 2018; Zhou et al., 2023). It generates diverse training scenarios by introducing visual or dynamic changes in simulators, with the hope that the unknown test environments can be covered. However, this approach has limitations: the training complexity scales with the number of variations, and insufficient randomization may lead to inadequate coverage of target domains. In contrast, target domain data is assumed to be accessible in domain adaptation (Zhu et al., 2023). RL-GAN employs unaligned GANs to translate target domains into familiar states in the source domain (Gamrian & Goldberg, 2019). CURL extracts high-level features from raw pixels using contrastive learning (Laskin et al., 2020b), and LUSR adopts Cycle-Consistent VAEs for unified state representation learning (Xing et al., 2021). Unlike these approaches, Lapse does not presume access to unknown target domains during deployment. We focus on adapting to evolvable state space via policy reuse.

**Policy Reuse**   Policy reuse learns multiple policies each specialised to their own subset of tasks so as to improve transfer to a new unseen task Fernández & Veloso (2006); Rosman et al. (2016); Li et al. (2019). One branch of methods train hierarchical high-level policies over source policies. Another branch of works aggregate source policies with the guidance of value functions on the target task Barreto et al. (2018); Cheng et al. (2020); Barekatain et al. (2021). (Barreto et al., 2018) focus on the situation where source tasks and target tasks share the same dynamics, and aggregate source policies by choosing the policy that has the largest Q value at each state based on successor features. CUP (Zhang et al., 2022) chooses a guidance policy that has the largest one-step improvement by querying the critic, and guides learning by regularizing the target policy to imitate the guidance policy. Bossens & Sobey (2024) adaptively assigns policies to tasks based on $\epsilon$-greedy bandit learning over lifetime reinforcement learning. Continual Reinforcement Learning (CRL) (Kessler et al., 2022; Li et al., 2024) focuses on adapting to new tasks quickly by reusing policy knowledge learned from old tasks, but they still require costly trial-and-error. All these approaches focus on efficiency on transfer to tasks with dynamics or reward gaps. We emphasize the evolution of state space and do not allow costly trial-and-error.

**Novelty Detection and Adaptation**   Focusing on the open-world decision making problem, a line of research works refer to abrupt changes in environments as novelty (Peng et al., 2021; Balloch et al., 2023). They propose to detect the novelty via knowledge graph or neuro-symbolic world model and achieve policy adaptation with imagination trajectories generated from the updated world model. These

methods fail to be directly applied to high-dimensional, continuous and complex environments, where dynamics and rewards are difficult to be depicted with simple rules. Zollicoffer et al. (2023) addresses this problem by extending it to latent-based novelty detection via DreamerV2 (Hafner et al., 2020) but does not study how to achieve policy adaptation in complex environments, mainly because the accuracy of imagination trajectories cannot be guaranteed. Our method takes a further step for the high-dimensional, continuous and complex environments via a novel solution.

**Learning with Evolvable Features** The concept of Feature Evolvable Streaming Learning (FESL) was introduced by Hou et al. (2017), focusing on an overlapping period where features from different stages are observable. OLVF (Beyazit et al., 2019) learns to classify the feature space and the instances simultaneously to handle arbitrarily varying spaces. FDESL (Zhang et al., 2020b) further extends the setting by assuming that the data distribution also changes. Our work is related to FESL paradigm, but they focus on online learning with data streams. We first consider the problem in RL and formulate it into SERL problem.

## 6. Conclusion

In this work, we present the **Lea**rning to reuse **p**olicies in **s**tate **e**volvable environments (**Lapse**) approach, addressing the challenge of evolvable state space in reinforcement learning (RL) with the formulation of state evolvable reinforcement learning (SERL). Lapse reuses policies from two aspects by combining a robust policy and an adaptive policy through state reconstruction and offline learning, thus avoiding the need for extensive and costly trial-and-error during deployment. Theoretical and empirical results on adaptation capability underscores Lapse 's potential in efficiently handling dynamic real-world scenarios. However, the requirement of $D_n$ can restrict the application scenarios in real world, and Lapse may be less effective in environments with drastic state space evolutions. Future work could focus on enhancing Lapse to handle more rapid state evolutions and exploring its applicability to more complex tasks. Another appealing direction is extending Lapse to multi-agent embodied intelligence settings (Feng et al., 2025), where multiple physically situated agents must coordinate and adapt in a jointly evolving state space. Furthermore, relaxing the need of behavioral experiences through Large Language Models (LLMs) (Kim et al., 2024) and human instructions is also a promising and valuable direction for future research.

## Acknowledgements

This work is supported by the National Science Foundation of China(62495093, U24A20324), the Natural Science Foundation of Jiangsu (BK2024119, BK20243039). We thank Tencent AI Arena for their support, and the anonymous reviewers for their support and helpful discussions on improving the paper.

## Impact Statement

This paper presents work whose goal is to advance the field of Reinforcement Learning (RL). Our method is intended to more effectively adapt RL policy to possible changes in state space during deployment. The work presented does not raise any additional ethical concerns, and thus no special discussion on ethical issues is required.

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

# A. Proofs for Theoretical Analysis

## A.1. Proof for Proposition 3.2

We first introduce the following lemma for the proof of Proposition 3.2.

**Lemma A.1.** *Let $f_n$ be an $\epsilon_R - \epsilon_P$ evolution of $\mathcal{M}_n$, which leads to $\mathcal{M}_{n+1}$. $f_n^{-1}$ is the inverse of the evolution. As is defined in Definition 2.1, $R_{n+1}(s_{n+1}, a) = \mathbb{E}_{s_n \sim f_n^{-1}(\cdot|s_{n+1})}[R_n(s_n, a)]$, and $\mathcal{P}_{n+1}(s'_{n+1}|s_{n+1}, a) = \mathbb{E}_{s_n, s'_n \sim f_n^{-1}(\cdot|s_{n+1}), f_n^{-1}(\cdot|s'_{n+1})}[\mathcal{P}_n(s'_n|s_n, a)]$. Then for any $s_{n+1} \in \mathcal{S}_{n+1}, a \in \mathcal{A}$,*

$$|\mathbb{E}_{s_n \sim f_n^{-1}(\cdot|s_{n+1})}[R_n(s_n, a)] - R_{n+1}(s_{n+1}, a)| \leq \epsilon_R.$$

*Proof.*

$$
\begin{aligned}
|\mathbb{E}_{s_n \sim f_n^{-1}(\cdot|s_{n+1})}[R_n(s_n, a)] - R_{n+1}(s_{n+1}, a)| &= |\sum_{s_n} f_n^{-1}(s_n|s_{n+1})[R_n(s_n, a)] - R_{n+1}(s_{n+1}, a)| \\
&\leq \sum_{s_n} f_n^{-1}(s_n|s_{n+1})|R_n(s_n, a) - R_{n+1}(s_{n+1}, a)| \\
&= \sum_{s_n} f_n^{-1}(s_n|s_{n+1})|R_n(s_n, a) - \sum_{\tilde{s}_n} f_n^{-1}(\tilde{s}_n|s_{n+1})R_n(\tilde{s}_n, a)| \\
&= \sum_{s_n} \mathbb{I}(f_n(s_{n+1}|s_n) > 0)f_n^{-1}(s_n|s_{n+1})|R_n(s_n, a) - \sum_{\tilde{s}_n} f_n^{-1}(\tilde{s}_n|s_{n+1})R_n(\tilde{s}_n, a)| \\
&\leq \sum_{s_n} \mathbb{I}(f_n(s_{n+1}|s_n) > 0)f_n^{-1}(s_n|s_{n+1})\epsilon_R \\
&= \epsilon_R
\end{aligned}
\tag{11}
$$

According to the bayesian inverse $f_n^{-1}(s_n|s_{n+1}) = \frac{f_n(s_{n+1}|s_n)}{\sum_{\tilde{s}_n} f_n(s_{n+1}|\tilde{s}_n)}$, $f_n(s_{n+1}|s_n) > 0$ only if $f_n^{-1}(s_n|s_{n+1}) > 0$. This allows for the application of Definition 3.1, which bounds the discrepancy of reward function when $f_n(s_{n+1}|s_n) > 0$.

It is also noticeable that we define $f_n^{-1}(s_n|s_{n+1}) = \frac{f_n(s_{n+1}|s_n)}{\sum_{\tilde{s}_n} f_n(s_{n+1}|\tilde{s}_n)}$ by assuming a uniform prior distribution $p_n(\cdot)$ over $\mathcal{S}_n$. We can also introduce occupancy measurement induced by uniform policy as prior, but it will not change the conclusion of the proposition. $\square$

Based on the above conclusion, we show that the performance gap between $\pi_n$ and $\pi_n \circ f_n^{-1}$ within $\mathcal{M}_n$ and $\mathcal{M}_{n+1}$ can be bounded by $\epsilon_R$ and $\epsilon_P$.

*Proof.* First of all, state value function $V_{\mathcal{M}}^\pi : \mathcal{S} \to \mathbb{R}$ is defined as $V_{\mathcal{M}}^\pi(s) := \mathbb{E}[\sum_{t=0}^\infty \gamma^t r_t | \pi, s_0 = s]$. Notice that, since $r_t \in [0, R_{\max}]$, we have

$$0 \leq V_{\mathcal{M}}^\pi(s) \leq \sum_{t=0}^\infty \gamma^t R_{\max} = \frac{R_{\max}}{1-\gamma}. \tag{12}$$

Furthermore, *Bellman policy operator* $\mathcal{T}_{\mathcal{M}}^\pi$ is defined as follows, when applied to some vector $f \in \mathbb{R}^{|\mathcal{S}|}$,

$$
\begin{aligned}
(\mathcal{T}_{\mathcal{M}}^\pi f)(s) &:= \mathbb{E}_{a \sim \pi(\cdot|s)}[R(s,a) + \gamma \mathbb{E}_{s' \sim \mathcal{P}(\cdot|s,a)}[f(s')]] \\
&= \mathbb{E}_{a \sim \pi(\cdot|s)}[R(s,a) + \gamma \langle \mathcal{P}(s,a), f \rangle].
\end{aligned}
\tag{13}
$$

Here $\langle \cdot, \cdot \rangle$ is dot product. The Bellman policy operator is a $\gamma$-contraction under $l_\infty$ norm (Puterman, 2014): for any $f, f' \in \mathbb{R}^{|\mathcal{S}|}$:

$$||\mathcal{T}_{\mathcal{M}}^\pi f - \mathcal{T}_{\mathcal{M}}^\pi f'||_\infty \leq \gamma ||f - f'||_\infty. \tag{14}$$

$V_{\mathcal{M}}^\pi$ is the fixed point of $\mathcal{T}_{\mathcal{M}}^\pi$, meaning $\mathcal{T}_{\mathcal{M}}^\pi V_{\mathcal{M}}^\pi = V_{\mathcal{M}}^\pi$.

Inspired by Jiang et al. (2015), we first prove that

$$||V_{\mathcal{M}_{n+1}}^{\pi_n \circ f_n^{-1}} - V_{\mathcal{M}_n}^{\pi_n} \circ f_n^{-1}||_\infty \leq \frac{\epsilon_R}{1-\gamma} + \frac{\gamma \epsilon_P R_{\max}}{2(1-\gamma)^2}. \tag{15}$$

Here, $(V_{\mathcal{M}_n}^{\pi_n} \circ f_n^{-1})(s_{n+1}) = \mathbb{E}_{s_n \sim f_n^{-1}(\cdot|s_{n+1})}[V_{\mathcal{M}_n}^{\pi_n}(s_n)]$ is defined over $\mathcal{S}_{n+1}$. To prove this, we first apply $\gamma$-contraction and fixed-point property of Bellman policy operator. For simplicity, let $\mathcal{T}_n^\pi := \mathcal{T}_{\mathcal{M}_n}^\pi$.

$$
\begin{aligned}
&||V_{\mathcal{M}_{n+1}}^{\pi_n \circ f_n^{-1}} - V_{\mathcal{M}_n}^{\pi_n} \circ f_n^{-1}||_\infty \\
&= ||(\mathcal{T}_{n+1}^{\pi_n \circ f_n^{-1}})V_{\mathcal{M}_{n+1}}^{\pi_n \circ f_n^{-1}} - (\mathcal{T}_{n+1}^{\pi_n \circ f_n^{-1}})(V_{\mathcal{M}_n}^{\pi_n} \circ f_n^{-1}) + (\mathcal{T}_{n+1}^{\pi_n \circ f_n^{-1}})(V_{\mathcal{M}_n}^{\pi_n} \circ f_n^{-1}) - V_{\mathcal{M}_n}^{\pi_n} \circ f_n^{-1}||_\infty \\
&\leq ||(\mathcal{T}_{n+1}^{\pi_n \circ f_n^{-1}})V_{\mathcal{M}_{n+1}}^{\pi_n \circ f_n^{-1}} - (\mathcal{T}_{n+1}^{\pi_n \circ f_n^{-1}})(V_{\mathcal{M}_n}^{\pi_n} \circ f_n^{-1})||_\infty + ||(\mathcal{T}_{n+1}^{\pi_n \circ f_n^{-1}})(V_{\mathcal{M}_n}^{\pi_n} \circ f_n^{-1}) - V_{\mathcal{M}_n}^{\pi_n} \circ f_n^{-1}||_\infty \\
&\leq \gamma ||V_{\mathcal{M}_{n+1}}^{\pi_n \circ f_n^{-1}} - V_{\mathcal{M}_n}^{\pi_n} \circ f_n^{-1}||_\infty + ||(\mathcal{T}_{n+1}^{\pi_n \circ f_n^{-1}})(V_{\mathcal{M}_n}^{\pi_n} \circ f_n^{-1}) - V_{\mathcal{M}_n}^{\pi_n} \circ f_n^{-1}||_\infty.
\end{aligned}
\tag{16}
$$

This implies that

$$
\begin{aligned}
||V_{\mathcal{M}_{n+1}}^{\pi_n \circ f_n^{-1}} - V_{\mathcal{M}_n}^{\pi_n} \circ f_n^{-1}||_\infty &\leq \frac{1}{1-\gamma} ||(\mathcal{T}_{n+1}^{\pi_n \circ f_n^{-1}})(V_{\mathcal{M}_n}^{\pi_n} \circ f_n^{-1}) - V_{\mathcal{M}_n}^{\pi_n} \circ f_n^{-1}||_\infty \\
&= \frac{1}{1-\gamma} ||(\mathcal{T}_{n+1}^{\pi_n \circ f_n^{-1}})(V_{\mathcal{M}_n}^{\pi_n} \circ f_n^{-1}) - (\mathcal{T}_n^{\pi_n} V_{\mathcal{M}_n}^{\pi_n}) \circ f_n^{-1}||_\infty.
\end{aligned}
\tag{17}
$$

For notation simplicity, let $R_n^\pi(s) := \mathbb{E}_{a \sim \pi(\cdot|s)} R(s,a)$, $\mathcal{P}_n^\pi(s) := \mathbb{E}_{a \sim \pi(\cdot|s)} \mathcal{P}_n(s,a)$, $a \sim \pi$ and $s_n \sim f_n^{-1}$ are abbreviation for $a \sim \pi(\cdot|s)$ and $s_n \sim f_n^{-1}(\cdot|s_{n+1})$. Then, for any $s_{n+1} \in \mathcal{S}_{n+1}$,

$$
\begin{aligned}
&|(\mathcal{T}_{n+1}^{\pi_n \circ f_n^{-1}})(V_{\mathcal{M}_n}^{\pi_n} \circ f_n^{-1})(s_{n+1}) - (\mathcal{T}_n^{\pi_n} V_{\mathcal{M}_n}^{\pi_n}) \circ f_n^{-1}(s_{n+1})| \\
&= |(\mathcal{T}_{n+1}^{\pi_n \circ f_n^{-1}})(V_{\mathcal{M}_n}^{\pi_n} \circ f_n^{-1})(s_{n+1}) - (\mathcal{T}_n^{\pi_n} V_{\mathcal{M}_n}^{\pi_n}) \circ (f_n^{-1})(s_{n+1})| \\
&= |R_{n+1}^{\pi_n \circ f_n^{-1}}(s_{n+1}) + \gamma \langle \mathcal{P}_{n+1}^{\pi_n \circ f_n^{-1}}(s_{n+1}), V_{\mathcal{M}_n}^{\pi_n} \circ f_n^{-1} \rangle - \mathbb{E}_{s_n \sim f_n^{-1}}[R_n^{\pi_n}(s_n)] - \gamma \langle \mathbb{E}_{s_n \sim f_n^{-1}}[\mathcal{P}_n^{\pi_n}(s_n)], V_{\mathcal{M}_n}^{\pi_n} \rangle| \\
&\leq \mathbb{E}_{a \sim \pi_{n+1}} |R_{n+1}(s_{n+1}, a) - \mathbb{E}_{s_n \sim f_n^{-1}}[R_n(s_n, a)]| + \gamma |\langle \mathcal{P}_{n+1}^{\pi_n \circ f_n^{-1}}(s_{n+1}), (\mathcal{F}_n^{-1})^\intercal V_{\mathcal{M}_n}^{\pi_n} \rangle - \langle \mathbb{E}_{s_n \sim f_n^{-1}}[\mathcal{P}_n^{\pi_n}(s_n)], V_{\mathcal{M}_n}^{\pi_n} \rangle| \\
&\leq \epsilon_R + \gamma |\langle \mathcal{F}_n^{-1} \mathcal{P}_{n+1}^{\pi_n \circ f_n^{-1}}(s_{n+1}), V_{\mathcal{M}_n}^{\pi_n} \rangle - \langle \mathbb{E}_{s_n \sim f_n^{-1}}[\mathcal{P}_n^{\pi_n}(s_n)], V_{\mathcal{M}_n}^{\pi_n} \rangle| \\
&= \epsilon_R + \gamma |\langle \mathcal{F}_n^{-1} \mathcal{P}_{n+1}^{\pi_n \circ f_n^{-1}} - \langle \mathbb{E}_{s_n \sim f_n^{-1}}[\mathcal{P}_n^{\pi_n}(s_n)], V_{\mathcal{M}_n}^{\pi_n} \rangle| \\
&= \epsilon_R + \gamma |\langle \mathcal{F}_n^{-1} \mathcal{P}_{n+1}^{\pi_n \circ f_n^{-1}} - \langle \mathbb{E}_{s_n \sim f_n^{-1}}[\mathcal{P}_n^{\pi_n}(s_n)], V_{\mathcal{M}_n}^{\pi_n} - \frac{R_{\max}}{2(1-\gamma)} \mathbf{1} \rangle| \\
&\leq \epsilon_R + \gamma \mathbb{E}_{a, s_n \sim f_n^{-1}} ||\mathcal{F}_n^{-1} \mathcal{P}_{n+1}(s_{n+1}, a) - \mathcal{P}_n(s_n, a)||_1 ||V_{\mathcal{M}_n}^{\pi_n} - \frac{R_{\max}}{2(1-\gamma)} \mathbf{1}||_\infty \\
&\leq \epsilon_R + \gamma \epsilon_P \frac{R_{\max}}{2(1-\gamma)}.
\end{aligned}
\tag{18}
$$

In the steps above, we expand bellman update by definition and use cauchy-schwarz inequality to split dot-product with $\Delta \mathcal{P}$ and value function. Lemma A.1 and $\epsilon_P$ consistency is applied to derive upper bound of the discrepancy of reward and transition functions.

Finally, according to the definition of the policy performance $J_n(\pi_n) := \mathbb{E}[\sum_{t=0}^\infty \gamma^t r_{n,t} | \pi_n, s_{n,0}, \mathcal{P}_n]$, $J_n(\pi_n) = V_{\mathcal{M}_n}^{\pi_n}(s_{n,0})$.

$$
\begin{aligned}
|J_{n+1}(\pi_n \circ f_n^{-1}) - J_n(\pi_n)| &= |V_{\mathcal{M}_{n+1}}^{\pi_n \circ f_n^{-1}}(s_{n+1,0}) - V_{\mathcal{M}_n}^{\pi_n}(s_{n,0})| \\
&= |V_{\mathcal{M}_{n+1}}^{\pi_n \circ f_n^{-1}}(s_{n+1,0}) - (V_{\mathcal{M}_n}^{\pi_n} \circ f_n^{-1})(s_{n+1,0})| \\
&\leq ||V_{\mathcal{M}_{n+1}}^{\pi_n \circ f_n^{-1}} - V_{\mathcal{M}_n}^{\pi_n} \circ f_n^{-1}||_\infty \\
&\leq \frac{\epsilon_R}{1-\gamma} + \frac{\gamma \epsilon_P R_{\max}}{2(1-\gamma)^2}.
\end{aligned}
\tag{19}
$$

$\square$

**Proof for Proposition 3.3**  We first introduce the following lemmas for the proof of Proposition 3.3

**Lemma A.2.** *With the definition of state occupancy* $d_\pi(s) = (1-\gamma)\sum_{t=0}^{\infty}\gamma^t Pr(s_t = s|\pi)$, *for any two different policies* $\pi^A, \pi^B$ *under the MDP* $\mathcal{M}$, *we have:*

$$D_{TV}(d_{\pi^A}, d_{\pi^B}) \leq \frac{\gamma}{1-\gamma}\mathbb{E}_{s\sim d_{\pi^A}}[D_{TV}(\pi^A(\cdot|s), \pi^B(\cdot|s))]. \tag{20}$$

*Proof.* We prove the lemma based on permutation theory presented in Schulman et al. (2015) and Xu et al. (2020). With $\mathcal{P}_\pi(s'|s) := \sum_{a\in\mathcal{A}}\mathcal{P}(s'|s,a)\pi(a|s)$ and $d_0$ specifies the initial state distribution, we have:

$$
\begin{aligned}
d_\pi &= (1-\gamma)\sum_{t=0}^{\infty}\gamma^t\mathrm{Pr}(s_t = s|\pi) \\
&= (1-\gamma)(\mathbb{I}-\gamma\mathcal{P}_\pi)^{-1}d_0.
\end{aligned} \tag{21}
$$

Let $G_A = (\mathbb{I}-\gamma\mathcal{P}_{\pi^A})^{-1}, G_B = (\mathbb{I}-\gamma\mathcal{P}_{\pi^B})^{-1}$, then we obtain:

$$
\begin{aligned}
G_A - G_B &= G_A(G_B^{-1} - G_A^{-1})G_B \\
&= G_A(\gamma\mathcal{P}_{\pi^A} - \gamma\mathcal{P}_{\pi^B})G_B \\
&= \gamma G_A(\mathcal{P}_{\pi^A} - \mathcal{P}_{\pi^B})G_B.
\end{aligned} \tag{22}
$$

With the equation, we have:

$$
\begin{aligned}
D_{\mathrm{TV}}(d_{\pi^A}, d_{\pi^B}) &= \frac{1}{2}\|d_{\pi^A} - d_{\pi^B}\|_1 \\
&= \frac{1}{2}\|(1-\gamma)(G_A - G_B)d_0\|_1 \\
&= \frac{1}{2}\|(1-\gamma)\gamma G_A(\mathcal{P}_{\pi^A} - \mathcal{P}_{\pi^B})G_B d_0\|_1 \\
&= \frac{1}{2}\|\gamma G_A(\mathcal{P}_{\pi^A} - \mathcal{P}_{\pi^B})d_{\pi^B}\|_1 \\
&\leq \frac{\gamma}{2}\|G_B\|_1\|\mathcal{P}_{\pi^A} - \mathcal{P}_{\pi^B})d_{\pi^A}\|_1.
\end{aligned} \tag{23}
$$

Here $G_B$ is bounded as:

$$
\begin{aligned}
\|G_B\|_1 &= \|(\mathbb{I}-\gamma\mathcal{P}_{\pi^B})^{-1}\|_1 \\
&= \left\|\sum_{t=0}^{\infty}\gamma^t P_{\pi^B}^t\right\|_1 \\
&\leq \sum_{t=0}^{\infty}\gamma^t\|P_{\pi^B}\|_1^t \\
&= \sum_{t=0}^{\infty}\gamma^t = \frac{1}{1-\gamma}.
\end{aligned} \tag{24}
$$

We can also show that:

$$
\begin{aligned}
\|(\mathcal{P}_{\pi^A} - \mathcal{P}_{\pi^B})d_{\pi^A}\|_1 &\leq \sum_{s,s'}|\mathcal{P}_{\pi^A}(s'|s) - \mathcal{P}_{\pi^B}(s'|s)|d_{\pi^A}(s) \\
&= \sum_{s,s'}\left|\sum_{a}\mathcal{P}(s'|s,a)(\pi^A(a|s) - \pi^B(a|s))\right|d_{\pi^A}(s) \\
&\leq \sum_{s,a,s'}\mathcal{P}(s'|s,a)|\pi^A(a|s) - \pi^B(a|s)|d_{\pi^A}(s) \\
&= \sum_{s}d_{\pi^A}(s)\sum_{a}|\pi^A(a|s) - \pi^B(a|s)| \\
&= 2\mathbb{E}_{s\sim d_{\pi^A}}[D_{\mathrm{TV}}(\pi^A(\cdot|s), \pi^B(\cdot|s))].
\end{aligned} \tag{25}
$$

We complete the proof by substituting Equation (24) and Equation (25) into Equation (23). $\qquad\square$

**Lemma A.3.** *With the definition of state-action occupancy $\rho_\pi(s,a) = \pi(a|s)d_\pi(s)$, for any two different policies $\pi^A, \pi^B$ under the MDP $\mathcal{M}$, we have:*

$$D_{TV}(\rho_{\pi^A}, \rho_{\pi^B}) \leq \frac{1}{1-\gamma} \mathbb{E}_{s \sim d_{\pi^A}}[D_{TV}(\pi^A(\cdot|s), \pi^B(\cdot|s))]. \tag{26}$$

*Proof.*

$$
\begin{aligned}
D_{\text{TV}}(\rho_{\pi^A}, \rho_{\pi^B}) &= \frac{1}{2} \sum_{(s,a)} |\rho_{\pi^A}(s,a) - \rho_{\pi^B}(s,a)| \\
&= \frac{1}{2} \sum_{s,a} |\pi^A(a|s)d_{\pi^A}(s) - \pi^B(a|s)d_{\pi^B}(s)| \\
&= \frac{1}{2} \sum_{s,a} |(\pi^B(a|s) - \pi^A(a|s))d_{\pi^A}(s) + (d_{\pi^B}(s) - d_{\pi^A}(s))\pi^B(a|s)| \\
&\leq \frac{1}{2} \sum_{s,a} |\pi^B(a|s) - \pi^A(a|s)|d_{\pi^A}(s) + \frac{1}{2} \sum_{s} |(d_{\pi^B}(s) - d_{\pi^A}(s))| \sum_{a} \pi^B(a|s) \\
&= \mathbb{E}_{s \sim d_{\pi^A}}[D_{\text{TV}}(\pi^A(\cdot|s), \pi^B(\cdot|s))] + D_{\text{TV}}(d_{\pi^A}, d_{\pi^B}) \\
&\leq \frac{1}{1-\gamma} \mathbb{E}_{s \sim d_{\pi^A}}[D_{\text{TV}}(\pi^A(\cdot|s), \pi^B(\cdot|s))].
\end{aligned}
\tag{27}
$$

We complete the proof with the last inequality following Lemma A.2. $\qquad\square$

Based on the above conclusions, we show that the performance gap can be bounded by divergence of two policies. For simplicity, we assume $f_n^{-1}$ and $g_n$ are deterministic and the conclusion holds by expanding it to the case of expectation.

*Proof.* Following Schulman et al. (2015), we can rewrite the expected return as:

$$
\begin{aligned}
J(\pi) &= \mathbb{E}\left[\sum_{t=0}^{\infty} R(s_t, a_t)\right] \\
&= \sum_{t=0}^{\infty} \gamma^t \sum_{s} \Pr(s_t = s|\pi) \sum_{a} \pi(a|s)R(s_t, a_t) \\
&= \sum_{t=0}^{\infty} \sum_{s,a} \rho_\pi(s,a)R(s_t, a_t) \\
&= \frac{1}{1-\gamma} \mathbb{E}_{\rho_\pi(s,a)}[R(s_t, a_t)].
\end{aligned}
\tag{28}
$$

Combining Prop. 3.2, Lemma A.3 and the condition, we have that:

$$
\begin{aligned}
&|J_{n+1}(\pi_n \circ f_n^{-1}) - J_{n+1}(\pi_n \circ g_n)| \\
&\leq \frac{1}{1-\gamma} \sum_{s^{n+1},a} |(\rho_{\pi_n \circ f_n^{-1}}(s^{n+1}, a) - \rho_{\pi_n \circ g_n}(s^{n+1}, a))R^{n+1}(s^{n+1}, a)| \\
&\leq \frac{2R_{\max}}{1-\gamma} D_{\text{TV}}(\rho_{\pi_n \circ f_n^{-1}}, \rho_{\pi_n \circ g_n}) \\
&\leq \frac{2R_{\max}}{(1-\gamma)^2} \mathbb{E}[D_{\text{TV}}((\pi_n \circ f_n^{-1})(\cdot|s^{n+1}), (\pi_n \circ g_n)(\cdot|s^{n+1}))] \\
&= \frac{2R_{\max}}{(1-\gamma)^2} \mathbb{E}[D_{\text{TV}}(\pi_n(\cdot|f_n^{-1}(s^{n+1})), \pi_n(\cdot|g_n(s^{n+1})))] \\
&\leq \frac{2R_{\max}}{(1-\gamma)^2} \eta.
\end{aligned}
\tag{29}
$$

Finally, combining Equation (29) and Proposition 3.2, we have

$$
\begin{aligned}
|J_n(\pi_n) - J_{n+1}(\pi_n \circ g_n)| =& |J_n(\pi_n) - J_{n+1}(\pi_n \circ f_n^{-1}) + J_{n+1}(\pi_n \circ f_n^{-1}) - J_{n+1}(\pi_n \circ g_n)| \\
\leq& |J_n(\pi_n) - J_{n+1}(\pi_n \circ f_n^{-1})| + |J_{n+1}(\pi_n \circ f_n^{-1}) - J_{n+1}(\pi_n \circ g_n)| \\
\leq& \frac{\epsilon_R}{1-\gamma} + \frac{\gamma \epsilon_P R_{\max}}{2(1-\gamma)^2} + \frac{2R_{\max}}{(1-\gamma)^2}\eta \\
=& \frac{\epsilon_R}{1-\gamma} + \frac{(\gamma \epsilon_P + 4\eta)R_{\max}}{2(1-\gamma)^2}.
\end{aligned}
\tag{30}
$$

This completes the proof. □

## B. Detailed Description of the Environments and Baselines

### B.1. Environments

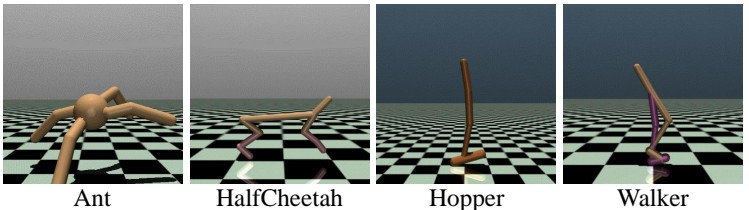

*Figure 6.* Mujoco environments used in this paper

**Gym Mujoco suite:** MuJoCo (Todorov et al., 2012) (Figure 6) is a high-fidelity physics engine designed for detailed and efficient rigid body simulations with contacts. It is widely used for benchmarking RL algorithms. Agents in MuJoCo environments receive vectorial state inputs and output continuous actions. For a comprehensive evaluation in our study, we have selected four tasks from the Gym MuJoCo suite:

- `Ant`: The Ant is a 3D robot consisting of one torso with four articulated legs, each with two links. The objective is to coordinate the movements of the legs to navigate towards a specified direction.

- `HalfCheetah`: The HalfCheetah is a 2-dimensional robot consisting of 9 links forming a spine, with 8 joints allowing articulation. The challenge is to exert torques on the joints to propel the cheetah forward as swiftly as possible.

- `Hopper`: The Hopper is a two-dimensional one-legged figure that consists of four main body parts - the torso at the top, the thigh in the middle, the leg in the bottom, and a single foot on which the entire body rests. The goal is to make hops that move in the forward direction.

- `Walker`: The Walker is a two-dimensional two-legged figure that consists of four main body parts - a single torso at the top, two thighs in the middle, two legs in the bottom, and two feet attached to the legs. The goal is to coordinate both sets of feet, legs, and thighs to move in the forward direction.

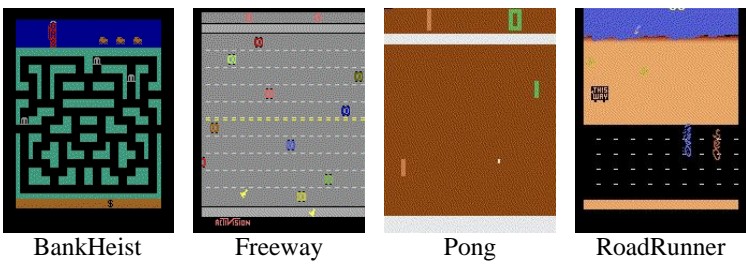

*Figure 7.* Atari environments used in this paper

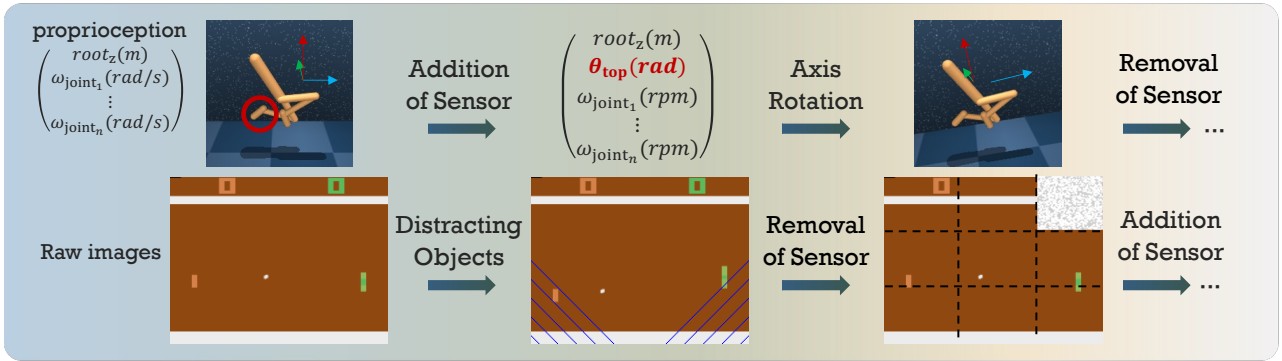

*Figure 8.* The state space evolution of experimental environments

**Atari games:** The Atari 2600 emulator (Figure 7) was introduced as an RL platform by Bellemare et al. (2013), offering a high-dimensional, pixel-based image input across a varied and complex set of tasks originally designed to challenge human players. We have selected four Atari games for comprehensive evaluation:

- `BankHeist`: A bank robber wants to rob as many banks as possible. The police chases him and will appear whenever he robs a bank. The robber controls a getaway car and must navigate maze-like cities while avoiding the caught of police. er the dynamite you have previously dropped.

- `Freeway`: The agent's objective is to guide the chicken across lane after lane of busy rush hour traffic.

- `Pong`: The agent controls the right paddle, and competes against the left paddle controlled by the computer. The agent each tries to keep deflecting the ball away from the right goal and into the left goal.

- `RoadRunner`: The road runner can walk in any direction and jump to outrun the opponent while avoiding the hazards of the desert.

## B.2. Evolutions

*Table 2.* Evolving functions applied in state space of Mujoco and Atari environments, respectively

|  | $f_0$ | $f_1$ | $f_2$ | $f_3$ | $f_4$ |
|---|---|---|---|---|---|
| Mujoco | remove sensors | rotate $(30 \pm 5)°$ | add sensors | rotate $-(30 \pm 5)°$ | linear mapping w/ noise |
| Atari | remove sensors | add moving objects | rotate $(30 \pm 5)°$ | add sensors | linear mapping w/ noise |

We here present the detailed evolutions applied in our experiments. The process is illustrated in Figure B.2 and specific setting of the evolutions are presented in Table 2. We evaluate the continual adaptation performance of different methods under various orders of evolutions.

## B.3. Baselines

- **RL-GAN** proposes the analogy-based zero-shot transfer via image-to-image translation. Specifically, it assumes access to unpaired data from the source and target domain beforehand, and achieves unaligned image-to-image translation using the Cycle-Consistency GANs. For our experiments, given access to paired data in dataset $D_n$, we implement RL-GAN using the Pix2Pix framework, which has shown better results than the original CycleGAN in our context.

- **LUSR** learns a latent unified state representation that is consistent across multiple domains. It applies Cycle-Consistent VAE ito disentangle the original state $s$ into a latent state representation composed of $\bar{s}$ and $\hat{s}$, which are domain-general and domain-specific, respectively. The policy will focus on the domain-general embeddings to make decisions. In the SERL problem, we learn disentangled latent representations and the offline policy using the collected datatset $D_n$.

- **PAD** makes the use of self-supervision to allow the policy to continue training after deployment. We decompose the policy into $\pi = (\pi_e, \pi_a, \pi_s)$, where we first have intermediate embedding $e_t = \pi_e(s_t)$ and makes decision based on $a_t \sim \pi_a(e_t)$. $\pi_s(\cdot, \cdot)$ takes $e_t$ and $e_{t+1}$ as input to predict the resulting action $a_t$. During the training stage, we optimize the RL and self-supervised objectives jointly to learn the policy. During the deployment stage, states are collected to optimize the self-supervised objective alone. In SERL problem, we learn the original policy via optimizing:

$$\mathcal{L}(\pi) = \mathcal{L}_{\text{SS}}(\pi_e, \pi_s) + \mathcal{L}_{\text{RL}}(\pi_e, \pi_a),$$
$$\mathcal{L}_{\text{SS}}(\pi_e, \pi_s) = l(a_t, \pi_s(\pi_e(s_t), \pi_e(s_{t+1}))),$$
(31)

where $l$ is the mean squared error between the ground truth and the model output in Mujoco and cross entropy in Atari. Then in evolving stage, we optimize the policy with the self-supervised objective $\mathcal{L}_{\text{SS}}$ alone.

- **Offline** learns a new policy for each $\mathcal{M}_{n+1}$ using data $D_n$ collected from the old policy. We choose sota offline RL methods: TD3+BC and CQL for Mujoco and Atari, respectively.

- **FPT** proposes a framework for Few-Shot Policy Transfer between two domains through State Mapping and Behavior Cloning. It uses the Cycle-Consistency GANs which aims at a map to clone the successful source task behavior policy to the target domain. We extend it to SERL problem by replacing Cycle-Consistency GANs with Pix2Pix for better alignment.

- **CUP** applies a critic-guided mechanism to reuse prior policies. It dynamically selects the policy with maximal one-step improvement at each state, forming a theoretically guaranteed guidance policy. The target policy is then regularized to imitate this guidance while preserving exploration. In our implementation, pretrained source policies guide the target policy before state evolution. The target policy is subsequently added to the source policy pool for future reuse.

## C. Algorithm and Experimental Details

In this section, we describe the detailed designs and techniques used for different modules of Lapse. We also provide the hyperparameters used in Lapse for consistency.

### C.1. Robust Training

Lapse requires a robust initial policy $\pi_0$ for efficient reuse and offline adaptive learning. We aim to improve the robustness by minimizing:

$$\mathcal{L}_{\text{robust}}^n = \mathbb{E}\left[\max_{\hat{s}^n \in T_n^\epsilon(s^n)} D_{\text{TV}}(\pi_n(\cdot|s^n), \pi_n(\cdot|\hat{s}^n))\right].$$
(32)

To deal with the intractable objective, we apply Wocar-PPO (Liang et al., 2022) and RADIAL-DQN (Oikarinen et al., 2021) for Mujoco and Atari, respectively.

**Wocar-PPO** is built on PPO (Schulman et al., 2017) which aims to optimize the policy with:

$$\mathcal{L}_{\text{RL}} = \mathbb{E}[-\min(\rho_t A(s_t, a_t), g_t A(s_t, a_t))]$$
(33)

where $\rho_t := \frac{\pi(a_t|s_t)}{\pi_{\text{old}}(a_t|s_t)}$ is importance sampling ratio, $g_t := \text{clip}(\rho_t, 1 - \eta, 1 + \eta)$, $\eta$ is a hyperparameter, and $A(s_t, a_t)$ is an estimate of the advantage function $A(s_t, a_t) := Q(s_t, a_t) - V(s_t)$. It also includes a term that minimizes the loss of the value function and rewards the entropy of the policy. Wocar-PPO introduces a worst-attack Bellman operator to estimate the lower bound of the policy value under the worst-case attack by minimizing the following estimation loss:

$$\mathcal{L}_{\text{est}} = \mathbb{E}[(\underline{Q}(s_t, a_t) - (r_t + \gamma \min_{\hat{a} \in \mathcal{A}_{adv}(s_{t+1}, \pi)} \underline{Q}(s_{t+1}, \hat{a})))^2],$$
(34)

where $\mathcal{A}_{adv}(s, \pi) := \{a \in \mathcal{A} | \exists \tilde{s} \in B^\epsilon(s) \text{ s.t. } \pi(\hat{s}) = a\}$. The worst action is selected through Interval Bound Propagation (IBP) (Wong & Kolter, 2018). With the worst-case value, it optimizes the policy to select actions that not only achieve high natural future reward, but also achieve high worst-case reward under adversarial attacks:

$$\mathcal{L}_{\text{wst}} = -\mathbb{E}[-\min(\rho_t \underline{Q}_t(s_t, a_t), g_t \underline{Q}_t(s_t, a_t))].$$
(35)

---

**Algorithm 1 `recon_train`$(n, D_n, \pi_n)$**

---

1: **Input:** current stage $n$, $\pi_n$
2: Initialize $g_n$ and $d_n$ which are parameterized by $\theta_{g_n}$ and $\theta_{d_n}$, respectively .
3: **for** $iter = 1$ **to** $T_{\max}$ **do**
4:    Sample a batch $\{(s_t^n, s_t^{n+1})\}$ from $D_n$.
5:    $\hat{s}_t^n \leftarrow g_n(s_t^{n+1})$.
6:    Compute $\mathcal{L}_{\text{GAN}}^{n+1}$ according to Equation (1).
7:    Optimize $d_n$ with $\theta_{d_n} \leftarrow \theta_{d_n} + \nabla_{\theta_{d_n}} \mathcal{L}_{\text{GAN}}^{n+1}$.
8:    Compute $\mathcal{L}_{\text{Lp}}^{n+1}$ and $\mathcal{L}_{\text{recon}}^{n+1}$ via Equations (2) and (3).
9:    Optimize $g_n$ with $\theta_{g_n} \leftarrow \theta_{g_n} - \nabla_{\theta_{g_n}} \mathcal{L}_{\text{recon}}^{n+1}$.
10: **end for**
11: Return $\pi_{n+1}^{\text{recon}} = \pi_n \circ g_n$.

---

Furthermore, it regularizes the policy network with a carefully designed state importance weight:

$$\mathcal{L}_{\text{reg}} = \mathbb{E}[w(s_t) \cdot \max_{\tilde{s}_t \in B^\epsilon(s_t)} \text{MSE}(\pi(s_t), \pi(\tilde{s}_t))], \tag{36}$$

where $w(s)$ is state importance weight defined as $w(s) := \max_{a_1 \in \mathcal{A}} Q(s, a_1) - \min_{a_2 \in \mathcal{A}} Q(s, a_2)$, which is approximated by $V(s_t) - \underline{Q}(s_t, a_t)$. Combining the above objectives together, we have the overall Wocar-PPO objective:

$$\mathcal{L}_{\text{Wocar}} = \mathcal{L}_{\text{RL}} + \kappa_{\text{wst}} \mathcal{L}_{\text{wst}} + \kappa_{\text{reg}} \mathcal{L}_{\text{reg}}, \tag{37}$$

where $\kappa_{\text{wst}}$ and $\kappa_{\text{reg}}$ are hyperparameters balancing between natural performance and robustness.

**RADIAL-DQN** is built upon Dueling-DQN, which splits Q-values into: $Q(s_t, a_t) = V_Q(s) + A_Q(s, a)$, and optimizes the Q network with the objective:

$$\mathcal{L}_{\text{RL}} = \mathbb{E}[(Q(s_t, a_t) - (r_t + \max_{a'} Q_{\text{tgt}}(s_{t+1}, a')))^2], \tag{38}$$

where $Q_{\text{tgt}}$ is a periodically updated target network. RADIAL-DQN designs a regularizer to minimize *overlap* between output bounds of actions with large difference in outcome. If there is no overlap, the original action's Q-value is guaranteed to be higher than others even under perturbation, so the agent won't change its behavior under perturbation. Meanwhile, it focuses on actions with different Q-values since taking a different but equally good action under perturbation is acceptable. RADIAL addresses it by adding a weight $Q_{\text{diff}}(s, a_1, a_2) := \max\{0, Q(s, a_2) - Q(s, a_1)\}$. Then, it defines $\bar{Q}(s, a, \epsilon) := \max_{\tilde{s} \in B^\epsilon(s)} Q(\tilde{s}, a)$ and $\underline{Q}(s, a, \epsilon) := \min_{\tilde{s} \in B^\epsilon(s)} Q(\tilde{s}, a)$, and the overlap between two actions: $Ovl(s, a_1, a_2, \epsilon) := \max(0, \bar{Q}(s, a_2, \epsilon)) - \underline{Q}(s, a_1, \epsilon) + \eta$, where $\eta := 0.5 \cdot Q_{\text{diff}}(s, a_1, a_2)$. The final loss function is as follows:

$$\mathcal{L}_{\text{adv}} = \mathbb{E}\left[\sum_{\hat{a}_t \in \mathcal{A}} Q_{\text{diff}}(s_t, \hat{a}_t, a_t) Ovl(s, \hat{a}_t, a_t)\right]. \tag{39}$$

Incorporating it with the original DQN loss, we have the overall RADIAL-DQN training objective:

$$\mathcal{L}_{\text{RADIAL}} = \kappa \mathcal{L}_{\text{RL}} + (1 - \kappa) \mathcal{L}_{\text{adv}}, \tag{40}$$

where $\kappa$ is a hyperparameter controlling the trade-off between standard performance and robust performance with value between 0 and 1.

## C.2. State Reconstruction Model

During the $n^{\text{th}}$ state evolution, we learn a reconstruction model mapping the new state space $S_{n+1}$ back to $S_n$ using collected paired data, so as to reuse the old policy $\pi_n$. We propose to use conditional GANs to achieve the state reconstruction as described in Alg. 1.

## C.3. Offline Adaptive Policy Learning

To learn a new policy which takes actions with new state space $\mathcal{M}_{n+1}$ during state space evolution, we choose TD3+BC for Mujoco and CQL for Atari. Considering that the policy is vulnerable to perturbations of inputs, we make use of a

conservative smoothing technique introduced by Yang et al. (2022) to make the policy network smoother. The policy will therefore be more robust and facilitate future reuse with the state reconstruction model. Specifically, we generate a bounded neighbouring area $B^\epsilon_{n+1}(s^{n+1}) = \{s \mid \|s - s^{n+1}\|_2 \le \epsilon, s \in \mathcal{S}_n\}$ around $s^{n+1}$. We simply denote it as $B^\epsilon$ in the following. Then compute the upper bound of the variation distance between $\pi^{\text{off}}_{n+1}(s_{n+1})$ and $\pi^{\text{off}}_{n+1}(\hat{s}_{n+1})$, $\hat{s}_{n+1} \in B^\epsilon$ according to Equation (5). We use mean square error as the variation distance metric so that the loss function is as follows:

$$\mathcal{L}^{n+1}_{\text{robust}} = \mathbb{E}\left[ \max_{\hat{s}^{n+1} \in B^\epsilon} \|\pi^{\text{off}}_{n+1}(\cdot|s^{n+1}), \pi^{\text{off}}_{n+1}(\cdot|\hat{s}^{n+1})\|^2_2 \right]. \tag{41}$$

---

**Algorithm 2 off_train$(n, D_n, \pi_n, \beta_{n+1})$**

1: Initialize a new policy $\pi^{\text{off}}_{n+1}$ and Q network $Q_{n+1}$, parameterized by $\phi_\pi$ and $\phi_Q$, respectively.
2: Set target parameters equal to $\phi_\pi$ and $\phi_Q$.
3: Compute the neighbouring area bound $\epsilon$ according to the scale of states in $D_n$.
4: **for** $iter = 1$ **to** $T_{\max}$ **do**
5:     Sample a batch $\{(s^{n+1}_t, a^{n+1}_t, r_t, s^{n+1}_{t+1})\}$ from $D_n$.
6:     Learn $Q_{n+1}$ and optimize $\phi_Q$ via the TD3 algorithm.
7:     Compute $\hat{\mathcal{L}}^{n+1}_{\text{off}}$ via Equation (6).
8:     **if** $iter$ **mod** $policy\_delay = 0$ **then**
9:         Generate a bounded neighbouring area $B^\epsilon(s^{n+1})$.
10:         Compute $\mathcal{L}^{n+1}_{\text{robust}}$ via Equation (41).
11:         Compute $\mathcal{L}^{n+1}_{\text{off}}$ via Equation (7).
12:         Optimize $\phi_\pi$ with $\phi_\pi \leftarrow \phi_\pi - \nabla_{\phi_\pi} \mathcal{L}^{n+1}_{\text{off}}$.
13:         Update target network with:

$$\phi_{targ,\pi} \leftarrow \tau_{\text{TD3}}\phi_{targ,\pi} + (1 - \tau_{\text{TD3}})\phi_\pi$$
$$\phi_{targ,Q} \leftarrow \tau_{\text{TD3}}\phi_{targ,Q} + (1 - \tau_{\text{TD3}})\phi_Q.$$

14:     **end if**
15: **end for**
16: Return $\pi^{\text{off}}_{n+1}$.

---

As for CQL applied in Atari games, the loss function is optimized as follows:

$$\hat{\mathcal{L}}^{n+1}_{\text{off}} = \beta_{n+1}\mathcal{L}_{\text{RL}} + \mathbb{E}\left[ \log \sum_a \exp Q(s_t, a) - Q(s_t, a_t) \right] \tag{42}$$

where $\mathcal{L}_{\text{RL}}$ is the standard DQN objective as is defined in Equation (38). We also implement it on Quantile Regression-DQN (Bellemare et al., 2017) for effectiveness. We schedule $\beta_{n+1} = \beta_{\max}(1 - \exp\{-\tau \cdot n\})$ as is defined in Section 4.2.

### C.4. Overall Training Algorithm

In SERL problem, we split it into the training phase and deployment phase. During the training phase, $\pi_0$ is allowed to be learned under $\mathcal{M}_0$ in an online manner. To improve the robustness, we apply Wocar-PPO and RADIAL-DQN into Mujoco and Atari environments, respectively, where the objective is described in Equation (37) and Equation (40). Before evolving into $\mathcal{M}_{n+1}$, we will learn the adaptive policy $\pi_{n+1}$ following the paradigm of Lapse as is introduced in Alg. 3. Meanwhile, for Atari games that only instantiate Q-function, we do not distinguish $\pi$ and $Q$ here and make decisions through $\arg\max_a \pi_{n+1}(a|s)$. Furthermore, to calculate the distance between two policies $\pi_1, \pi_2$, we approximate it through $\mathbb{E}[D_{\text{KL}}(\pi_1(\cdot|s), \pi_2(\cdot|s))]$.

---

**Algorithm 3 `Lapse`**

---

1: **Input:** Environment `Env`, current stage $n \geq 0$, old policy $\pi_n$, size of dataset $B_n$
2: Schedule the coefficient for offline adaptive policy learning: $\beta_{n+1} \leftarrow \beta_{\max}(1 - \exp(-\tau \cdot n))$.
3: Collect $D_n = \{(s_t^n, s_t^{n+1}, a_t, r_t, s_{t+1}^n, s_{t+1}^{n+1})\}$ from `Env` within $B_n$ episodes using $\pi_n$.
4: $\pi_{n+1}^{\text{recon}} \leftarrow \textbf{recon\_train}(n, D_n, \pi_n)$.
5: $\pi_{n+1}^{\text{off}} \leftarrow \textbf{off\_train}(n, D_n, \pi_n, \beta_{n+1})$.
6: Compute the distance $D(\pi_n, \pi_{n+1}^{\text{recon}})$ and $D(\pi_n, \pi_{n+1}^{\text{off}})$.
7: Compute $\kappa_{n+1}$ according to Equation (9).
8: $\pi_{n+1} \leftarrow \kappa_{n+1} \pi_{n+1}^{\text{recon}} + (1 - \kappa_{n+1}) \pi_{n+1}^{\text{off}}$.
9: $\pi_{n+1} \leftarrow \textbf{Pruning}(\pi_{n+1})$. {(Optional)}
10: Return $\pi_{n+1}$.

---

## C.5. The Hyperparameter Choice of Lapse

As $\pi_0$ is trained following the paradigm of Wocar-PPO and RADIAL-DQN, default parameters in the frameworks are used. We introduce the hyperparameter choices of the left parts in Table 3 and omit the subscript $n + 1$ for simplicity.

*Table 3.* Hyperparameter choices of Lapse

| Env | Hyperparameter | Value |
|---|---|---|
| Mujoco | p value in $\mathcal{L}_{\text{Lp}}$ | 2 |
| | $\lambda$ in $\mathcal{L}_{\text{recon}}$ | 10 |
| | $\beta_{\max}$ in $\hat{\mathcal{L}}_{\text{off}}$ | 2.5 |
| | $\tau$ in $\hat{\mathcal{L}}_{\text{off}}$ | 0.5 |
| | $\alpha^{\text{robust}}$ in $\hat{\mathcal{L}}_{\text{off}}$ | 0.1 |
| | $\epsilon$ in $B^{\epsilon}$ | 0.001 |
| | policy update interval | 2 |
| | $T_{\max}^{\text{recon}}$ | 10000 |
| | $T_{\max}^{\text{off}}$ | 10000 |
| | learning rate | $3e-4$ |
| | target update $\tau_{\text{TD3}}$ | 0.005 |
| | $\gamma$ | 0.99 |
| Atari | p value in $\mathcal{L}_{\text{Lp}}$ | 1 |
| | $\lambda$ in $\mathcal{L}_{\text{recon}}$ | 10 |
| | $\beta_{\max}$ in $\hat{\mathcal{L}}_{\text{off}}$ | 0.1 |
| | $\tau$ in $\hat{\mathcal{L}}_{\text{off}}$ | $\infty$ |
| | $\alpha^{\text{robust}}$ in $\hat{\mathcal{L}}_{\text{off}}$ | 0.1 |
| | $\epsilon$ in $B^{\epsilon}$ | $3/255$ |
| | ema coefficient | 0.995 |
| | ema interval | 100 |
| | $T_{\max}^{\text{recon}}$ | 20000 |
| | $T_{\max}^{\text{off}}$ | 40000 |
| | learning rate | $3e-4$ |
| | number of quantiles in QRDQN | 200 |
| | target update interval | 400 |
| | $\gamma$ | 0.99 |

# D. Additional Experimental Results

## D.1. Complete Adaptation Performance

Figure 10 illustrates the compelete continuous adaptation of various methods to the evolvable state space by comparing the performance over 5 different stages. The result is consistent with the conclusion in the main text. Despite the increased performance degradation over successive evolving stages, Lapse maintains superior performance compared with all other baseline methods.

## D.2. Complete Training Curves of Lapse

We present the complete training curves in Figure 11, where blue dashed lines represent the performance during new paired data collection. Note that the policy is deployed to the environment only after training for the current stage is complete, thereby preventing costly trial-and-error during adaptation.

## D.3. Pruning Strategies

We define $\pi_{n+1} = \kappa_n \pi_{n+1}^{\text{recon}} + (1 - \kappa_n)\pi_{n+1}^{\text{off}}$ in a recursive way due to the reuse of old policies in $\pi_{n+1}^{\text{recon}}$. To demonstrate the contribution of each model, we introduce occupancy weights $c_{0:n+1}$ defined as:

$$c_i = \begin{cases} \prod_{i=1}^{n} \kappa_i, & i = 0 \\ (1 - \kappa_i)\prod_{j=i+1}^{n+1} \kappa_j, & i = 1, 2, ..., n \\ 1 - \kappa_{n+1}, & i = n + 1 \end{cases} \tag{43}$$

Then we can expand the expression as:

$$\pi_{n+1} = c_0 \pi_0 \circ \hat{g}_0 + \sum_{i=1}^{n} c_i \pi_i^{\text{off}} \circ \hat{g}_i + c_n \pi_{n+1}^{\text{off}}, \tag{44}$$

where $\hat{g}_i = g_{i+1} \circ g_{i+2} \circ \cdots \circ g_{n+1}$. This requires us to store all the learned reconstruction models and the offline adaptive policies, which is unbearable in memory storage and inference delay with the increase of the evolving stages. Take Mujoco environments Ant and Walker as examples, not all components of Lapse $\pi_5$ are important as depicted in Figure 9. This way we can just use $\pi_{4,5}^{\text{off}}, \pi_5^{\text{recon}}$ in Ant and $\pi_{0,4,5}^{\text{off}}, \pi_{1:5}^{\text{recon}}$ in Walker. The occupancy weight of one model will not grow in the future by definition, so that we can only store the models that contribute a lot. To mitigate the problem, we propose to prune the heavy compound policies based on the decomposed non-recursive expression of $\pi_{n+1}$ as shown in Equation (44).

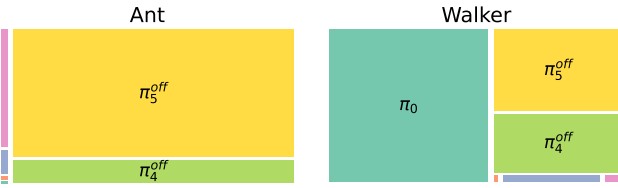

*Figure 9.* The occupancy weights of Lapse policy in Mujoco

---

**Algorithm 4** `Pruning`$(\pi_{n+1})$

---

1: Decompose and compute the occupancy weights $\{(c_i)\}$ via Equation (43).
2: **for** $i = 0$ to $n + 1$ **do**
3:    **if** $c_i < pruning\_threshold$ **then**
4:       $c_i \leftarrow 0$. {Ignore the model that contributes few}
5:    **end if**
6: **end for**
7: Normalize the occupancy weights:

$$c_i' \leftarrow \frac{c_i}{\sum_{j=0}^{n+1} c_j}, i = 0, 1, ..., n + 1.$$

8: Return the pruned Lapse policy with $\{(c_i')\}$ via Equation (44).

---

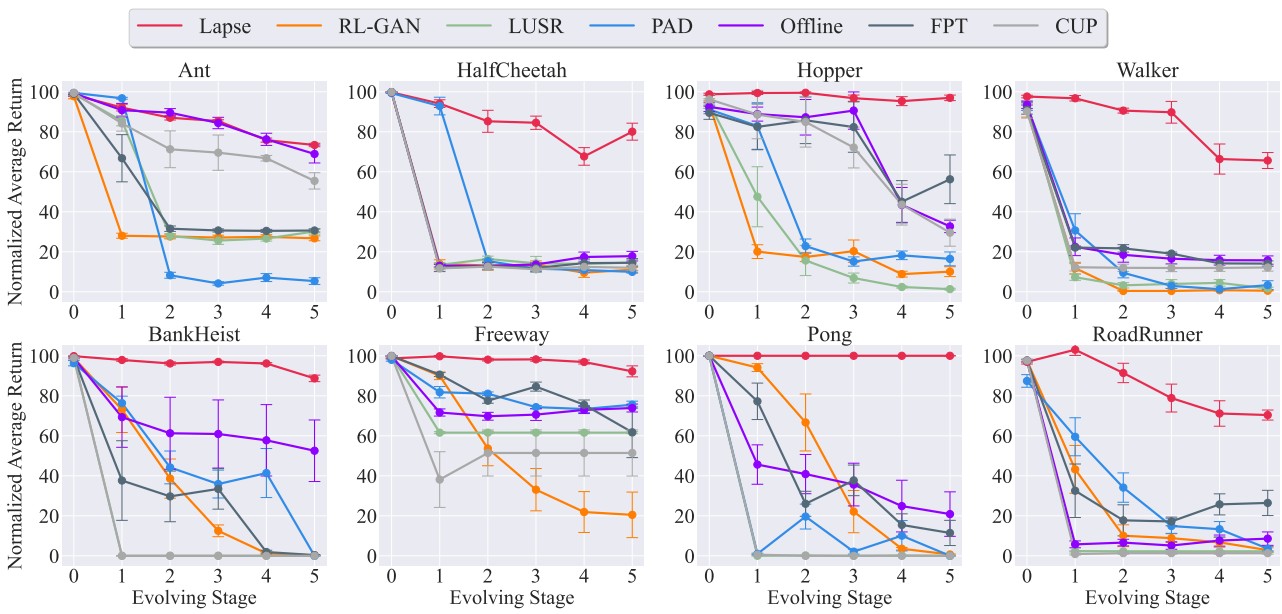

*Figure 10.* The adaptation performance of the learned policy as the state space evolves

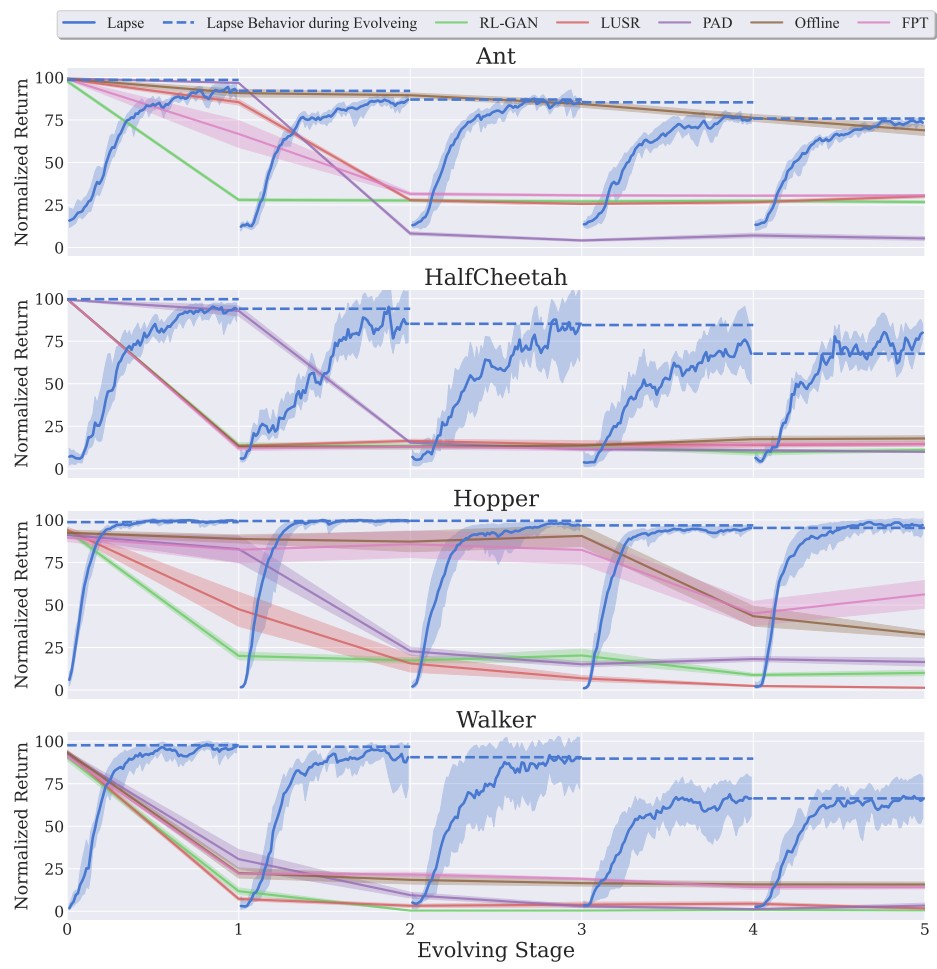

*Figure 11.* Complete training curves of Lapse

## D.4. Ablation Studies

**Ablation on Different Components**    We dissect the impact of each component within Lapse as depicted in Figure 12. *W/o offline* and *W/o recon* break down on different tasks, illustrating that neither of them can handle continuous adaptation problems alone. *W/o robust*'s poor performance on 3 out 4 tasks again emphasizes the necessity of the robustness. *W/o kappa* shows inferior adaptation capability compared to Lapse, manifesting the benefit of the automatic ensemble weight adjustment. To analyze the contribution of each component, we conducted a series of experiments in which components were incrementally added, culminating in the complete version of Lapse. Figure 13 shows this ablation study, demonstrating the impact of each added component and verifying their necessity.

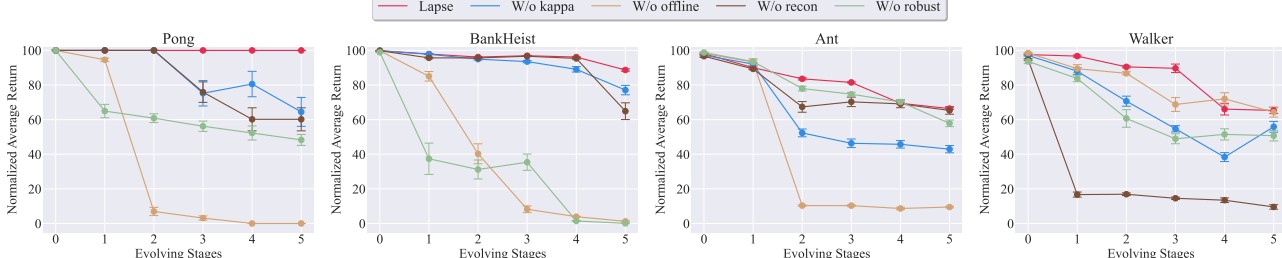

*Figure 12.* The overall ablation studies

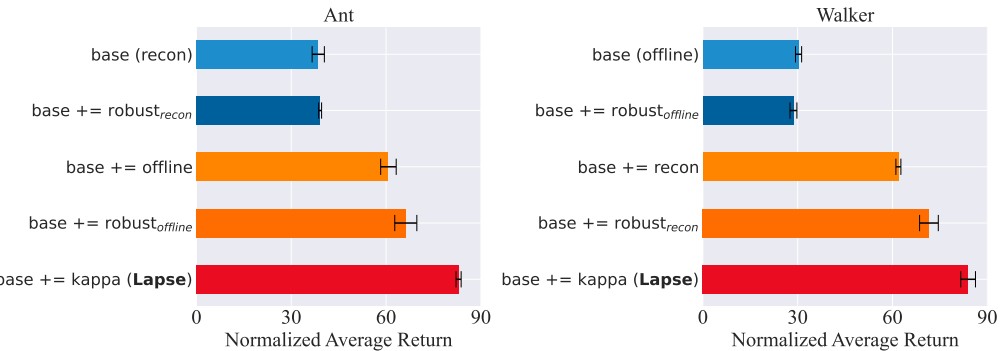

*Figure 13.* Demonstration of each component's contribution

**Ablation on Ensemble Methods**    We employ linear mixing because determining weights via Equation (9) and performing linear mixing are cost-free and highly efficient. We evaluated how various ensemble methods affect the performance of Lapse. The results appear in Figure 14. We do not plot the learning curve for stacking ensembles, as their returns never exceeded 1000 even after more than one million training steps. Alternative methods, such as hypernetworks, also showed no significant improvement.

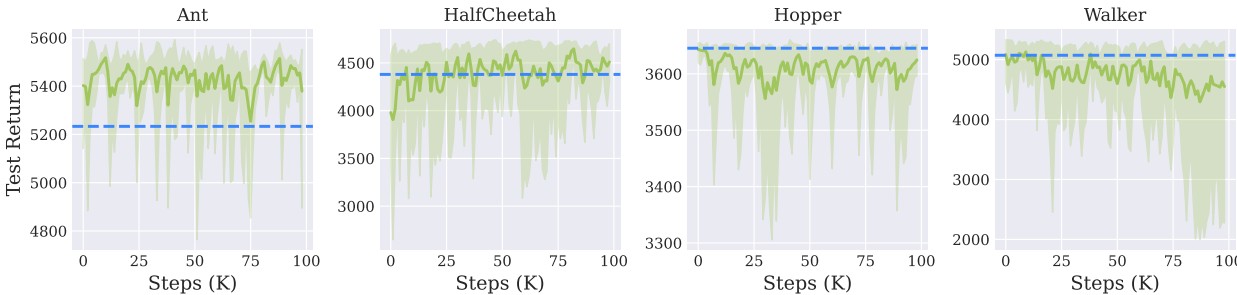

*Figure 14.* Test performance of dynamic ensemble weighting based on a hyper-network. The dashed line represents the performance achieved by linear mixing using the kappa from Lapse, while the solid curve demonstrates performance when dynamically learning ensemble weights via online RL using a hyper-network, which takes the state as input and outputs a weight vector used to weight each action dimension for ensemble aggregation.

*Table 4.* Comparison of average performance $R$, storage $N$ and inference delay $T$ between vanilla Lapse and pruning strategies across 2 tasks. Results are averaged over 5 distinct seeds. The storage metrics are measured by thousands of parameters of $\pi_i$ and the inference delay metrics are measured by the number of seconds it takes for $\pi_i$ to perform $10^3$ inferences in our experimental environment.

| | | $\mathcal{M}_0$ | $\mathcal{M}_1$ | $\mathcal{M}_2$ | $\mathcal{M}_3$ | $\mathcal{M}_4$ | $\mathcal{M}_5$ |
|---|---|---|---|---|---|---|---|
| | | Ant | | | | | |
| $R$ | vanilla | 100.00 | 101.70 | 93.63 | 86.51 | 80.62 | 69.27 |
| | pruning | 100.00 | 92.38 | 94.94 | 90.97 | 83.52 | 77.91 |
| $N$ | vanilla | 40.98 | 156.04 | 271.10 | 386.17 | 501.23 | 616.29 |
| | pruning | 40.98 | 156.04 | 133.61 | 133.61 | 133.61 | 133.61 |
| $T$ | vanilla | 0.36 | 0.68 | 1.01 | 1.33 | 1.66 | 1.98 |
| | pruning | 0.36 | 0.68 | 0.54 | 0.54 | 0.54 | 0.54 |
| | | Walker | | | | | |
| $R$ | vanilla | 100.00 | 98.70 | 102.17 | 69.33 | 75.36 | 70.74 |
| | pruning | 100.00 | 95.62 | 87.90 | 86.71 | 76.41 | 78.32 |
| $N$ | vanilla | 5.71 | 84.07 | 162.43 | 240.78 | 319.14 | 397.50 |
| | pruning | 5.71 | 12.12 | 18.54 | 24.96 | 31.38 | 61.77 |
| $T$ | vanilla | 0.14 | 0.46 | 0.78 | 1.10 | 1.43 | 1.75 |
| | pruning | 0.14 | 0.28 | 0.43 | 0.57 | 0.71 | 0.91 |

**Ablation on Pruning Strategies**   We compare average performance, storage and inference delay metrics between vanilla Lapse and pruning strategies with $pruning\_threshold = 0.2$ across 2 tasks. As shown in Table 4, we find that our pruning strategies save a lot of storage space and inference delay while ensuring excellent performance. In Ant, the pruning strategy only retains $\pi_i^{\text{off}}$ of the latest stage $i$ when $i > 1$, which means that storage and inference delay remain lightweight and fast. In Walker, the pruning strategy only retains $\pi_0$ and all $\pi_i^{\text{recon}}$, which saves most of the storage and nearly half inference delay. The above experiments show our pruning strategies maintain SERL scalability and performance.

### D.5. Sensitivity Studies

We further conduct more experiments in Mujoco to investigate how hyperparameters influence the performance of Lapse. The results can be seen in Figure 15. We selected a set of hyperparameters that performed well across tasks.

Additionally, in order to explore the impact of the size of $D_n$ on our method, we conduct experiments with size ranging from 6 to 18 trajectories. As shown in Figure 16, our method is robust to the various size, which is beneficial to obtain stable and good performance during the deployment stage.

### D.6. More evolving stages

To provide a more comprehensive evaluation of Lapse's adaptation performance over an extended number of evolving stages, we expanded the experiment to include 15 stages—three times the number in our initial tests. These additional stages were simulated using scalar multiplication, with pruning strategies employed to maintain scalability.

Figure 17 showcases the results of this extended evaluation. Lapse demonstrates commendable adaptation performance in three of the four tested environments. However, in the Walker task, Lapse's performance declines after the 7[th] stage. This drop in performance may be related to the inherent offline learning inefficiencies of TD3+BC within the Walker environment, as noted by Fujimoto & Gu (2021). It is anticipated that substituting TD3+BC with a more efficient algorithm could preserve Lapse's adaptability in this scenario.

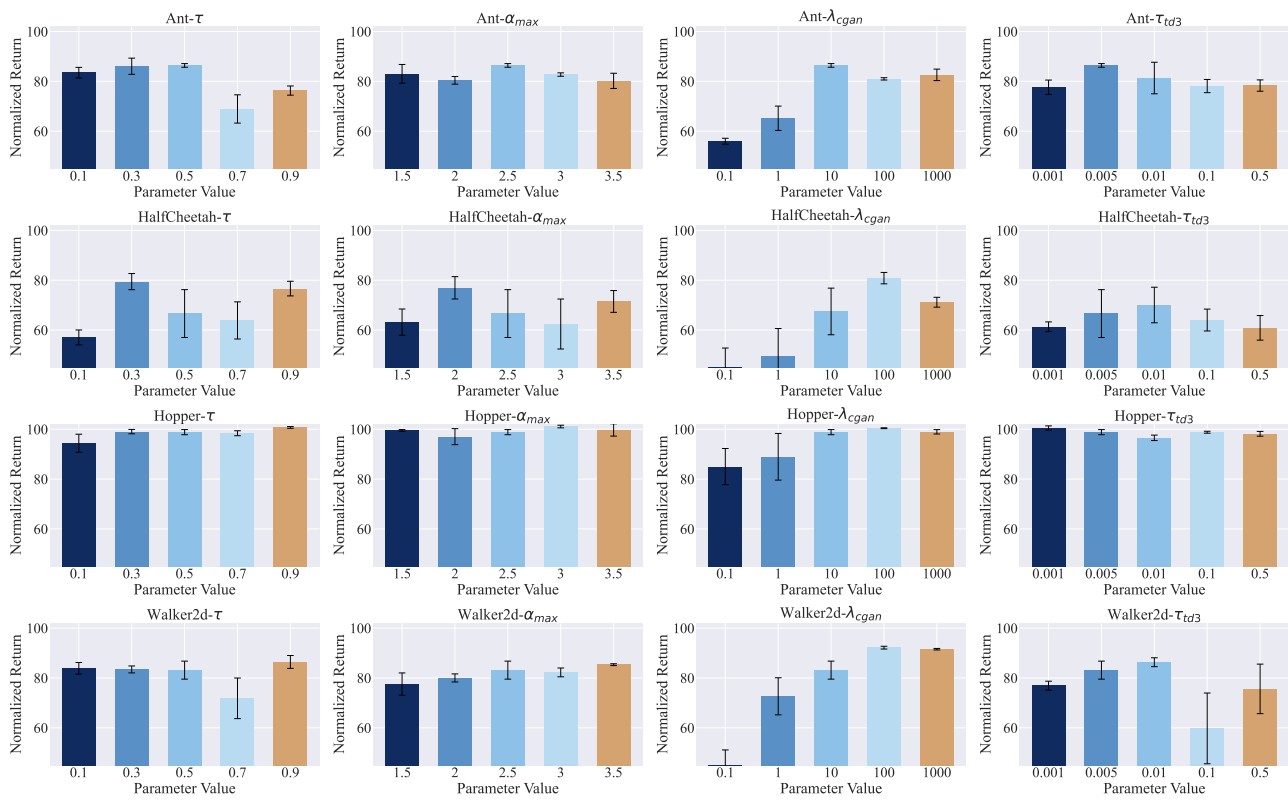

Figure 15. The overall parameter sensitivity studies

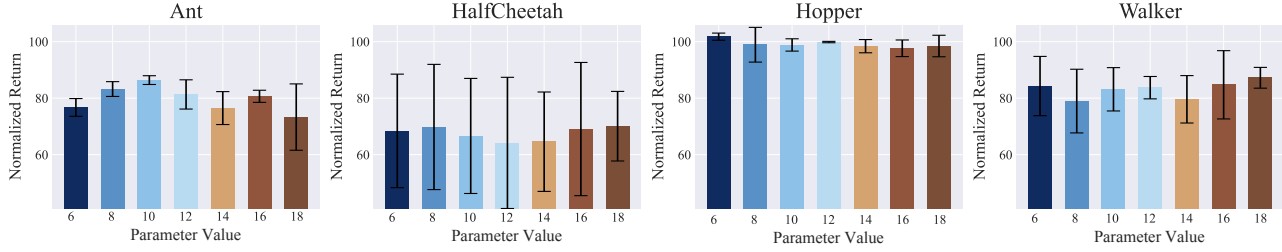

Figure 16. Sensitivity studies of the size of datatset $D_n$

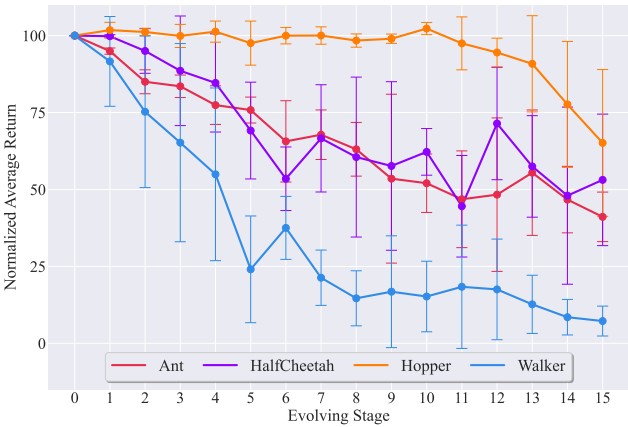

Figure 17. Continual adaptation under more evolving stages

