# OpenReview forum: "Learning to Reuse Policies in State Evolvable Environments"
_ICML.cc/2025/Conference — ICML 2025 poster_

### Official Review · Reviewer_9wCY · 2025-03-12

**Overall Recommendation:** 3

**Summary:**

This work introduces introduces a framework to handle non-stationary state distributions during deployment of an RL agent. The authors are motivated by the deterioration or maintenance of sensors on real-world systems that do not have a constant fidelity. The proposed framework is structured around a State Evolvable RL problem where there is a series of MDPs $\mathcal{M}_n$ in which the progression is defined by $f_n$. The agent has access to a series of datasets of the evolved data. The agent attempts to learn a policy that can reuse previous behavior as well as adapt to new state distributions. The framework proposes a combination of robust RL algorithms and explicit learning of the state evolution through reconstruction. The authors provide theoretical analysis and extensive experiments in Mujoco (state-based) and Atari (pixel-based) tasks.

**Claims And Evidence:**

The main claim in the paper is that the Lapse method retains better performance through an evolving state space across domains. This claim is supported in the Atari and Mujoco domains in Table 1. The experiments were run over several seeds, and standard deviation is shown. The baselines are discussed in the appendix, but a nuanced understanding of why the baselines do not perform well compared to Lapse cannot be garnered from the paper. Having said that,  in the appendix the authors address more classic methods for training robust agents, such as Domain Randomization.

**Essential References Not Discussed:**

n/a

**Experimental Designs Or Analyses:**

The majority of experimental design in shown in Appendix section C. The main paper does not include how the state evolution actually progresses in the experiments, but this information is present in the appendix. Figure 2 and Table 1 demonstrate the effectiveness of the Lapse. In my experience with Mujoco, a small change in state variable can have large impact on the resultant policy; the changes evolutions that the authors design are reasonable.

**Methods And Evaluation Criteria:**

The main method is a combination of state evolution modeling, policy regularization, and adaptive policy aggregation.
## Methods
The state evolution modeling is learned through classic GAN-style losses and is reasonable in this setting (Equations 2, 3). The policy regularization for robustness is from Equation 4 and includes a standard TV distance; typically, to regularize to a policy people often use KL divergence and I wonder why these authors opted for TV Distance.

For offline policy learning, the authors invoke TD3+BC (Fujimoto 2021), which is a method that balances optimism and dataset regularization. I do not understand why the authors opted for an offline RL method when the task is naturally online. Perhaps an off-policy method such as SAC would be more well-suited.

For adaptive policy aggregation, the authors do linear mixing of the robust policy and the offline RL policy according to a a weight defined in Equation 9. The linear mixing of the policies makes strict assumptions about the policy structure and an action's influence on the environment. It is not always the case that linearly mixing the policies will work; for instance, when the action distributions don't have joint support.

**Other Comments Or Suggestions:**

Some nitpicks:


- the paper uses the phrase "on the other hand" 6 times in the paper.
- The writing in general is somewhat awkward and at times hinders a smooth reading process.
  - e.g. "The policy trained by reinforcement learning makes decisions based on state features collect from sensors"
  -
- Section 3.3, betwen lines 203 and 204, the word should be "specifically", not "In specific"

**Other Strengths And Weaknesses:**

In general, this paper is well-motivated and describes a general framework for a real-world problem.

### Weaknesses
- Assumes that changes due to state-evolution can be mapped iteratively to the original state distribution. I believe this is a restrictive assumption. The authors make claims that they can deal with the deterioration of the iterated mapping by adaptively favoring more recent data. It would be ideal if the framework could continuously learn a robust policy or be able to map directly from evolution n to evolution 0.
- Requires paired data between evolutions. It is not clear to me when it is reasonable to make this assumption, if ever. Would be more general if the framework could deal automatically pair data based on some estimation of which distribution is came from.
- Requires knowledge of when the state is evolving. It seems more likely that the evolution is a slow, and continuous process rather than a fast, discrete one. I recognize that maintenance is discrete and fast, but sensor deterioration is not.
- The distinction between state features and state space is not clear.

**Questions For Authors:**

In addition to the concerns I voiced in the above sections, I have several explicit questions.

1. Why opt for offline RL instead of off-policy RL for the behavior knowledge compenent?
2. In general, where is the new data coming from at each stage of the evolution? Why not use an online algorithm to more quickly adapt? It seems that the method would incur significant cost when collecting new data in the evolved state space?
3. Can you distinguish between evolving state-space, state features, and observation space?

**Relation To Broader Scientific Literature:**

This paper develops a framework for a real-world problem in life long learning. Learning robust policies for lifelong deployment and fine-tuning has been of interest in recent years and this paper fits well into that space. Due to the many components of Lapse, it might make the framework brittle to changes in the real world, and so this work may require testing on real-world data to assess its efficacy.

**Theoretical Claims:**

The theoretical claims (Section 3.4) are minimal in the main paper, but provide bounds on the return of the policy under bounded conditions of the MDP modeling.

---

> ### Author Rebuttal · Authors · 2025-04-01
>
> Thank you for the constructive comments and suggestions. We provide our responses below and present the additional experimental results in https://anonymous.4open.science/r/Lapse-7846.
>
> ### Q1. Performance of baselines
>
> The baselines underperform because they cannot anticipate all possible state evolutions during training. Instead, they must adapt to new state space using only a small set of paired transitions collected by $\pi_n$.  In general, domain adaptation methods require a large amount of data to extract generic representations for adaptation. CUP cannot enumerate all possible evolutions beforehand, leading to failure for direct reuse of source policies. We will expand this analysis in Sec. 4.2.
>
> ### Q2. Linear Mixing
>
> Linear mixing is an efficient ensemble method, which has been applied in policy ensemble [1, 2]. We employ linear mixing because determining the weight via Eqn. 9 and performing linear mixing is cost-free and highly-efficient. We conduct experiments on how the way of ensemble affects the performance of Lapse.  Alternatives like hypernetworks showed no significant improvement (**Figure 1**). We also train stacking but it failed in 4 tasks (score < 1000 after 1M steps).
>
> ### Q3 TV divergence
>
> Since $D_{TV}^2\leq \frac{1}{2}D_{KL}$, both divergences can achieve robustness. On the other hand, the loss is derived from Prop. 3.3, where performance gap is bounded by TV divergence. As discussed in Appx. D.1, the methods we use [4] do not directly optimize over TV divergence but take more efficient ways.
>
> ### Q4 New data, cost and algorithms
>
> In SERL formulation, the policy is deployed after training on the initial MDP $M_0$. During deployment, the state space may evolve for various reasons such as scheduled maintenance. Access to the information is motivated by the fact that real-world systems undergo scheduled sensor checks or replacements. For instance, if a sensor is nearing the end of its lifespan, a replacement can be installed in advance, allowing the agent to observe features from both old and new state spaces. The new dataset $D_n$ is collected by $\pi_n$ utilizing only state features $s_n$ from the old state space, but paired states $s_{n+1}$ can also be reserved.
>
> Our method will not incur significant cost as the policy for data collection is $\pi_n$ which performs well in $M_n$. Avoiding costly trial-and-error during policy adaptation is the primary contribution of Lapse.
>
> Since $\pi_{n+1}^{off}$ is learned using data collected by $\pi_n$, we choose offline RL algorithms. Online RL including off-policy algrorithms is not suitable because trial-and-error is not allowed in SERL.
>
> ### Q5 Continuously learn a robust policy or direct map to evolution 0
>
> When the state space evolves to $ S_{n+1}$, the depolyed policy $\pi_{n+1}$ is composed of both $\pi_{n+1}^{recon}$ and $\pi_{n+1}^{offline}$, where the latter does not require state reconstruction and maintains robustness via regularization.
>
> In Appx. E.2, we employ pruning strategies and verify that the initial policy can be omitted via proper pruning in some scenarios, without the need for iterative map. Meanwhile, we conduct experiments by mapping evolution n to 0 via cycleGAN. **Figure 2** shows our method performs better than directly aligning in most circumstances as it can better utilize the paired data.
>
> ### Q6 Paired data and possible enhancement
>
> Despite the limitations in some scenarios, it is still applicable for many real-world systems. Please refer to our response in **Q2 to Reviewer Uqfd for details** due to word limit.
>
> The idea that automatically pairing data is a valuable direction for further research. Cycle-GAN [6] is widely-used for domain adaptation by introducing cycle consistency loss. It implicitly pairs data from different domains. However, as the experimental analysis presents in **Figure 2**, it fails to achieve satisfying adaptation performance with direct application. In future works, developing relevant methods for automatically paring using a small set of data will help extend our method to more general scenarios.
>
> ### Q7 State features, space and observation space
>
> In our work, state space is the fundamental element $\mathcal S_n$ in the MDP $\mathcal M_n$, while state features $s_n \in \mathcal S_n$ are the input to the policy for decision making. The change of sensors lead to the evolution of state space, together with state features. There is no distinguishment between “observation” and “state” because we do not consider the partially observable setting. We are sorry for confusion caused by some abuse of these concepts. We will replace observation into state for consistency and clarify two concepts in the revised version.
>
> ### Q8 Nitpicks
>
> Thank you for pointing out the issues.
>
> [1] Towards Applicable RL: Improving the Generalization and Sample Efficiency with Policy Ensemble
>
> [2] Seerl: Sample efficient ensemble RL
>
> [3] Efficient adversarial training without attacking: Worst-case-aware robust RL

---

### Official Review · Reviewer_cpBN · 2025-03-17

**Overall Recommendation:** 3

**Summary:**

The authors propose the framework Learning to reuse policies in state evolution (Lapse) to solve the issue of model degradation when there is an explicit change in the environment representation that is received by the agent in the form of input. They formalize this scenario as State Evolvable Reinforcement Learning (SERL): a RL environment where there is a sudden change in visual representation. In the majority of RL algorithms, the model would face an extreme degradation in performance when the input representation changes like this, but Lapse avoids this by training a robust RL policy and using the rollouts from this policy as an offline dataset to train a new policy. The old policy and the new policy are then aggregated together to create a new policy for the evolved environment. The authors demonstrate strong results against a number of baselines, demonstrating the ability of their proposed method.

**Claims And Evidence:**

The claims in the paper are supported by proof (Section 3.4) or experimental results (4.2). All results were run and reported over a sufficient number of seeds.

**Essential References Not Discussed:**

I do believe there is a non-trivial body of work by Mark Riedl's group (Specifically Jonathan Balloch and Xiangyu Peng) in Novelty Detection and Policy reuse that the authors have missed. Notably, how the authors of the former frame a 'novelty' seems extremely similar to the authors definition of an SERL: a "sudden change in visual or state transitions".  In the case of the latter, their work with roleplaying agents also touches on reusing policies to adapt to a new environment (in their case, a new character roleplaying 'persona').

**Experimental Designs Or Analyses:**

I have checked whether the baselines evaluated against made sense in the context of the author's primary research question, if the experiments were run over enough seeds to be statistically reliable, and if the evolutions within the environment were fair and cherry-picked. I have found no issues with any of the previously mentioned

**Methods And Evaluation Criteria:**

The primary issue the authors seek to address with their method is the degradation of model performance as a result of changes to the underlying state of the environment. Their approach is a three-step process where overfitting to learned features in the current environment is actively penalized; the now more malleable policy is used to gather offline trajectories in the new/evolved environment; and the offline trajectories are then used to train a new policy for the evolved environment that is aggregated with the old one for the final policy.

I believe this process to be well motivated for the issue the authors are investigating. I do have a question regarding the final step that I will ask later in the review.

**Other Comments Or Suggestions:**

N/A

**Other Strengths And Weaknesses:**

Strengths:

1.) The authors research question is well-motivated and grounded in a very real and serious issue: RL algorithms would not be deployable in the real world if they catastrophically deviated from their policy at state representation perturbations.
2.) The authors use a straight-forward but well-motivated method (and valid) method for forcing a policy to not be overfit to the current state representation.
3.) They rigorously prove the effectiveness of the state reconstruction model
4.) They provide a thorough analysis on each baseline used in their evaluation and ensure that the changes in the environment are fair to all evaluated baselines

Weaknesses:
1.) It appears the change in state must be directly signaled to the algorithm for it to begin the adaptation process.
2.) The missing related work mentioned above.

**Questions For Authors:**

1.) Does Lapse require for the change in the environment to be directly signal to it? My impression was yes but please correct me if I am wrong.

2.) I believe the works mentioned in the "Essential References not Discussed" merit a reference, if not a slightly longer discussion on the differences between 'evolution' and 'novelty'. However, I am open to being convinced otherwise by the authors.

**Relation To Broader Scientific Literature:**

The author's work is relevant to the Reinforcement Learning community as an exploration in how to create robust and rapidly adapting policies in a changing environment. It is also relevant to the larger Machine Learning community as a whole.

**Theoretical Claims:**

I checked the correctness of equations 1-10, along with the definitions and propositions in section 3.4. I have also checked the expanded proof for Proposition 3.2 in the appendix (Section A) along with training details in Section D. I have found no issues with any of these.

---

> ### Author Rebuttal · Authors · 2025-04-01
>
> Thank you for careful review our paper and providing constructive comments and suggestions. We offer some clarification to your questions here, and we would appreciate any further comments you might have.
>
> ### Q1 Changes are directly signaled to Lapse
>
> Yes, the change in the environment should be directly signaled in our current SERL problem formulation. Access to sudch information is motivated by the fact that many real-world systems undergo scheduled sensor checks, replacements, or recalibrations, where developers can prepare beforehand. For example, large-scale industrial systems often have planned maintenance schedules  for updating or replacing sensors [1]. These scenarios allow developers to prepare for changes in advance, making our signaling assumption reasonable for practical deployment.
>
> ### Q2 Lack of essential references
>
> Thank you for bringing the references to our attention. The works referenced by the reviewer [2, 3, 4] and our Lapse all focus on the open-world/environment decision making problem. The sudden changes of environments are defined as “novelty” in these works, which emphasizes the abruptness of the change, while the “evolution” of state space in Lapse, although still unpredictable before deployment, can be scheduled beforehand for a **short** period. Specifically, [2, 3] detect the sudden change via knowledge graph or neuro-symbolic world model and achieve policy adaptation with imagination trajectories generated from the updated world model. These methods fail to be directly applied to high-dimensional, continuous and complex environments, where dynamics and rewards are difficult to be depicted with simple rules. [5] addresses this problem by extending it to latent-based novelty detection via DreamerV2 but does not study how to achieve policy adaptation in complex environments, mainly because the accuracy of imagination trajectories cannot be guaranteed. Our method takes a further step for the high-dimensional, continuous and complex environments via a novel solution, and we hope it could fill in gaps for open-environment.
>
> Aside from the works mentioned by the reviewer, some other works focus on dealing with sudden changes in decision making, involving environmental dynamics [5] and teammate in multi-agent systems [6]. We will add the corresponding discussions to the paper in the revised version. Thank you again for pointing out the problem.
>
> [1] A Methodology for Online Sensor Recalibration
>
> [2] Detecting and Adapting to Novelty in Games
>
> [3] Neuro-Symbolic World Models for Adapting to Open World Novelty
>
> [4] Novelty Detection in Reinforcement Learning with World Models
>
> [5] Adapt to Environment Sudden Changes by Learning a Context Sensitive Policy
>
> [6] Fast Teammate Adaptation in the Presence of Sudden Policy Change

---

### Official Review · Reviewer_Uqfd · 2025-03-21

**Overall Recommendation:** 3

**Summary:**

This paper introduces the problem of state evolvable reinforcement learning (SERL) and proposes a method called Lapse to address it.
The key contributions are as follows:
* Formalization of the SERL problem, where the state space of an environment evolves over time due to sensor changes.
* Lapse takes a two-pronged approach to policy reuse. On one hand, it directly reuses old policies via state reconstruction to handle vanishing sensors. On the other hand, it reuses behavioral knowledge through offline RL to better utilize new sensors.
* The method also incorporates automatic adjustment of ensemble weights to combine the two policy reuse approaches effectively.
* Additionally, the authors provide a theoretical analysis of performance gaps caused by state evolution uncertainty.
* Empirically, Lapse demonstrates significant performance improvements.

**Claims And Evidence:**

The claims made in the paper appear to be generally supported by clear evidence. The formalization of SERL is mathematically sound, providing a solid foundation. The Lapse method is described in detail, clearly explaining its components. The theoretical analysis justifies the approach and provides bounds on performance gaps. Empirical results show substantial performance improvements over baselines.

However, some claims need further scrutiny. The assumption of access to maintenance information seems strong, and the paper needs more motivation on why this specific setting is important. Without strong motivation, the applicability of the method to real-world scenarios is unclear. The limitation of considering only state evolution without changes to dynamics is also a significant constraint, potentially limiting the generalizability of the approach.

Furthermore, the lack of evaluation in a real-world environment where state evolution occurs as expected weakens the overall evidence for the method's practical utility. These limitations impact the strength of the claims made regarding the method's effectiveness in real-world SERL problems.

**Essential References Not Discussed:**

This is a well-written paper and most of the related works section covers the essential background and research papers in this area.

**Experimental Designs Or Analyses:**

The experimental design seems reasonably well-structured. Using both MuJoCo (vector based) and Atari (image-based) environments provides a diverse set of test cases, covering different state representations. Testing the policy in environments with multiple unknown state space evolutions evaluates the method's adaptation capability. Comparison against existing adaptation and transfer approaches provides context for the claimed performance improvements.

However, the absence of a compute analysis compared to the baselines is a significant omission. Without information on computational costs, it is difficult to assess the practical feasibility of the proposed method. Moreover, the limited presentation of training curves for all environments makes it challenging to fully evaluate the method's performance and stability across different settings. These limitations hinder a complete assessment of the experimental results.

**Methods And Evaluation Criteria:**

The proposed methods, in general, appear sensible for the SERL problem. Using GANs for state reconstruction is a reasonable approach to handle vanishing sensors, addressing a key challenge. Incorporating robustness regularization addresses potential issues arising from reconstruction errors, enhancing the method's resilience. Leveraging offline RL to learn from limited experience in new state spaces is a clever way to adapt to new sensors, making efficient use of available data.

The evaluation criteria using MuJoCo and Atari environments are standard benchmarks in RL research. These environments seem appropriate for demonstrating the effectiveness of the proposed method across different types of state spaces, including both vectorial and image-based representations. The baselines are also chosen appropriately.

**Other Comments Or Suggestions:**

* More details on practical implications and potential real-world applications of SERL would be helpful.
* A discussion of computational requirements and scalability issues would be valuable. Including a real-world example would greatly enhance the paper's impact.
* Adding training curves for all environments would provide a more comprehensive view of performance.
* A compute analysis comparing Lapse to baselines would offer insights into efficiency and scalability.

**Other Strengths And Weaknesses:**

Strengths:

* Novel problem formulation (SERL) that addresses a practical challenge in deploying RL systems.

* Creative combination of multiple techniques (GANs, robust RL, offline RL) to address the problem.

* Both theoretical analysis and empirical evaluation are provided.

Weaknesses:

* The assumption of prior knowledge about sensor changes may limit applicability in some scenarios.

* The assumption of access to maintenance information seems strong, and the paper needs more motivation on why this exact setting is important.

* The limitation of only considering state evolution without changes to dynamics/reward structures is a constraint.

* Lack of evaluation in a real-world environment where state evolution occurs naturally.

* Absence of a compute analysis compared to the baselines.

* Limited presentation of training curves for all environments.

* There should be another figure after Figure 10 that demonstrates what the addition of each component does for a complete ablation study.

**Questions For Authors:**

* Can you provide more motivation for assuming access to maintenance information and why this specific setting is important?
* Have you considered extending the method to handle changes in dynamics?
* Are there plans to evaluate Lapse in a real-world environment?

**Relation To Broader Scientific Literature:**

The paper positions its work well in relation to several areas of RL research. It builds on work in policy reuse/selection and adaptation to new MDPs. It leverages ideas from robust RL to address uncertainties in state reconstruction and incorporates offline RL techniques for learning from limited experience.

**Theoretical Claims:**

The paper includes theoretical analysis in Section 3.4. I did check the Propositions 3.2 and 3.2 and their detailed proofs in Appendix A. The high-level reasoning about bounding performance gaps based on evolution uncertainty and reconstruction errors appears sound.

---

> ### Author Rebuttal · Authors · 2025-04-01
>
> Thank you for your careful review and thoughtful comments. We hope our responses can relieve your concern. We present the additional experimental results in https://anonymous.4open.science/r/Lapse-13F8.
>
> ### Q1: Prior knowledge about sensor changes
>
> While this assumption does not hold universally, it is practical in many real-world systems. For instance: Wear-and-tear or battery depletion in industrial sensors is often predictable, allowing preemptive deployment of replacements.
>
> Meanwhile, the assumption of accessing a large amount of test-domain data is common in domain adaptation [1]. In our work, the prior knowledge about sensor changes only offers a small set of paired test-domain samples.
>
> ### Q2: Access to maintenance information
>
> Access to maintenance information is motivated by the fact that many real-world systems undergo scheduled sensor checks, replacements, or recalibrations [2]. The developers are thus allowed to capture paired data before sensor changes, which is essential for our method to adapt policies effectively with minimal online exploration. For example, large-scale industrial systems often have planned maintenance schedules to update or replace sensors [3]. The system should keep operating during the aforementioned periods.
>
> The problem pointed by the reviewer is valuable as there are many abrupt changes in the real world, which is also pointed out as the limitation of Lapse in conclusion part. Some works have also focused on dealing with sudden changes in dynamics [4]. However, these methods are only applicable for in-distribution changes and fail to cope with totally unseen scenarios. Online fine-tuning based adaptation is an alternative direction without the need of maintenance information, but it inevitably introduces costly trial-and-error. In future work, combining Lapse with these methods by restricting the policy adaptation in a safe and trustworthy region is of promise.
>
> ### Q3: Changes of dynamics and rewards
>
> Our work mainly focuses on the continual evolution of state space, which is built on SERL formulation. Paired states cannot be created if dynamics/rewards change, so we fail to directly apply Lapse to such scenarios. However, to investigate the potential of Lapse, we simply extend it to a more complex multi-agent scenario, SMAC, by combining it with a multi-task MARL method, MaskMA [5]. As the paired transitions do not exist, we assume access to a limited offline dataset when encountering new tasks. **Figure 1** demonstrates Lapse's potential for adaptation to unseen tasks when combined with multi-task/meta learning methods.
>
> ### Q4: Real-world experiments
>
> To further investigate how Lapse performs in real-world applications, we here choose Highway-env as alternatives for real-world applications including autonomous driving. As shown in **Figure 2**,  Lapse consistently outperforms other baselines in adaptation performance.  For more realistic physical embodied scenarios involving robotics, we leave it for future work.
>
> ### Q5: Lack of complete training curves, compute analysis and ablation study
>
> 1. We list out the complete training curves in the **Figure 3**, where blue dashed lines represent the performance when performing new paired data collection. It should be noticed that the policy is deployed to the environment for the next evolving stage until finishing the training, thus preventing costly trial-and-error during adaptation.
> 2. We present the compute analysis during training in **Table 1** by computing the number of parameters required for training and FLOPs during each stage. Our method demonstrates better performance with no significant extra computation cost.
> 3. The ablation study that demonstrates what the addition of each component does is presented in **Figure 4**, verifying the necessity of each component.
>
> [1] A Comprehensive Survey on Source-free Domain Adaptation
>
> [2] Open-world machine learning: applications, challenges, and opportunities
>
> [3] A Methodology for Online Sensor Recalibration
>
> [4] Adapt to Environment Sudden Changes by Learning a Context Sensitive Policy
>
> [5] MaskMA: Towards Zero-Shot Multi-Agent Decision Making

---

### Decision · Program_Chairs · 2025-05-01

**Decision:**

Accept (poster)

**Comment:**

This paper introduces State Evolvable Reinforcement Learning (SERL), a novel and practical problem formulation addressing the challenge of deploying RL agents in environments where state representations change over time. The proposed Lapse method cleverly combines GANs for state reconstruction, robust RL policies, and offline RL. The authors provide strong theoretical justification for their approach, including performance bounds, and demonstrate substantial empirical improvements. Reviewers highlight the well-motivated methods, sound experimental design, and the paper's relevance to policy reuse and adaptation literature.
Addressing reviewers' concerns, the authors have now included additional experiments in Highway-env, complete training curves, ablation studies, and compute analysis, as well as clarifying the assumptions regarding access to maintenance information. Given these significant improvements and the paper's strong contributions, I recommend acceptance and encourage the authors to fully integrate all proposed changes and address all remaining reviewer comments in the final version.